# Broadscale dampening of uncertainty adjustment in the aging brain

Julian Q. Kosciessa [1,2,3] ✉, Ulrich Mayr [4], Ulman Lindenberger[1,2] & Douglas D. Garrett [1,2] ✉

The ability to prioritize among input features according to relevance enables adaptive behaviors across the human lifespan. However, relevance often remains ambiguous, and such uncertainty increases demands for dynamic control. While both cognitive stability and flexibility decline during healthy ageing, it is unknown whether aging alters how uncertainty impacts perception and decision-making, and if so, via which neural mechanisms. Here, we assess uncertainty adjustment across the adult lifespan ($N = 100$; cross-sectional) via behavioral modeling and a theoretically informed set of EEG-, fMRI-, and pupil-based signatures. On the group level, older adults show a broad dampening of uncertainty adjustment relative to younger adults. At the individual level, older individuals whose modulation more closely resembled that of younger adults also exhibit better maintenance of cognitive control. Our results highlight neural mechanisms whose maintenance plausibly enables flexible task-set, perception, and decision computations across the adult lifespan.

Prioritizing goal-relevant input features is central to cognitive control and adaptive behaviors. But how do we discern relevant signals from distractions? While some contexts explicitly highlight specific features (e.g., a single road sign emphasizing school children)[1], most contexts provide only sparse (e.g., a "!" sign) or contrasting cues (e.g., multiple signs: school children, bicycles, construction, ...). Whereas selective cues enable us to prioritize individual features with high acuity, ambiguity about which input features are goal-relevant (i.e., *task uncertainty*) demands broader levels of sensitivity, even at the expense of precision[2,3]. An adaptive system should track the moment-to-moment variations in uncertainty, and tune perception, guide decisions, and select actions accordingly[4,5]. Here, we examine whether a failure to adapt computations to varying task uncertainty is a key characteristic of healthy human cognitive aging.

Behavioral observations support aging-related deficits in uncertainty adjustment. In contexts that cue specific task-relevant features of compound stimuli, older adults remain sensitive also to irrelevant features[6,7], indicating challenges in stable feature selection[8–11].

Conversely, older adults show inflexibility when contexts require dynamic feature switches[12–14], and incur substantial "fade-out" costs when transitioning from dynamic to stable contexts[15]. Such observations suggest that older adults may be stuck in a suboptimal 'middle ground' that neither affords stable task selectivity when uncertainty is low, nor task flexibility in dynamic or uncertain contexts. Although age-related deficits have been reported for aligning computations (e.g., learning rate) to uncertainty[16], it remains unclear whether such underutilization arises from challenges in estimating uncertainty, or from an inability to leverage adequate estimates. For uncertainty to provide a principled and comprehensive lens on aging-related adaptivity constraints, first evidence is required to establish whether and/or how neural responses to uncertainty differ in the older adult brain.

How brain function adjusts to variable uncertainty remains debated[17], but emerging models implicate interacting systems that define task sets, tune perception, and inform decision formation[18–20]. Task-set management has been localized to fronto-parietal cortex[20,21], with recent evidence suggesting additional thalamic contributions in

[1]Max Planck UCL Centre for Computational Psychiatry and Ageing Research, Berlin, Germany. [2]Center for Lifespan Psychology, Max Planck Institute for Human Development, Berlin, Germany. [3]Radboud University, Donders Institute for Brain, Cognition and Behaviour, Nijmegen, The Netherlands. [4]Department of Psychology, University of Oregon, Eugene, OR, USA. ✉e-mail: kosciessa@mpib-berlin.mpg.de; garrett@mpib-berlin.mpg.de

uncertain contexts[22,23]. When task sets specify target features, perceptual networks can constrain relevant information by combining distractor inhibition[24] with target enhancement[25]. In contrast, high uncertainty about goal-relevant targets may facilitate sensitivity to multiple features via broad increases in excitability[26]. Such regime switches can be orchestrated by diffuse neurotransmitter systems that adjust computational precision to changing demands[2]; for example, pupil dilation (as a proxy)[27] transiently increases alongside uncertainty[28,29]. In young adults, we observed such an integrated response to rising uncertainty[30], encompassing increased fronto-thalamic BOLD activation, increased pupil diameter, and increased EEG-based cortical excitability. These results indicate that interacting systems enable adaptive responses to variable task uncertainty. But does the responsiveness of these systems differ across the adult lifespan?

Initial behavioral evidence from reward-learning paradigms suggests that older adults are less able to represent and use uncertainty[16]. Moreover, the general observation that older adults' brain activity is less responsive to varying demands[31–33] is suggestive of less adaptive responses per se. Senescence is marked by changes across multiple systems, including diminished prefrontal cortex function[34], metabolic decreases in cognitive control networks[35–37], progressive deterioration of subcortical neurotransmitter systems[38–40] alongside reduced pupil size modulation[41], reduced cortical inhibition[42,43], and structural declines of coordinating nodes such as the thalamus[44,45]. Many of these systems can be linked to the representation of, and adaptive response to, uncertainty[30]. Yet, there is also a long-standing challenge in the cognitive neuroscience of aging to identify, and distinguish between, competing functional explanations for changes in adaptivity. Reductions in working-memory capacity[46], inhibition[47], or processing speed[48] have all been proposed as general changes underlying a wide range of deficits. The fact that age differences usually occur even in minimal-demand baseline conditions[32] can additionally complicate inferences from observed age differences in adaptivity. Here, we use convergent evidence from a broad spectrum of behavioral and neural signatures (decision modeling, EEG, fMRI, pupillometry) alongside a host of controls to establish altered uncertainty processing as a core feature of human brain aging.

Here, we tested whether we could explain individual differences in adaptivity among older adults. Specifically, a "maintenance account of aging"[49] suggests that cognitive deficits with senescence emerge when neural resources become insufficient to meet demands, which implies that older adults with neural engagement resembling that of younger adults should better maintain function. We test this account by examining the degree to which older adults express a young-adult pattern of specific neuro-behavioral signatures when adapting to uncertainty.

In this work, we examined multimodal signatures (decision modeling, EEG, fMRI, pupillometry) in 47 younger (avg. 26 years) and 53 older (avg. 69 years) adults to comprehensively test uncertainty adjustment across the adult lifespan. Participants performed a perceptual decision task that manipulated uncertainty about which feature(s) of a compound stimulus would become decision relevant. By assessing signatures that change under task uncertainty in younger adults'[30], we highlight dampened uncertainty modulation in older adults along with more constrained changes to perceptual evidence integration. Older adults with brain responses more closely resembling younger adults showed benefits in feature selection, providing initial evidence that maintained uncertainty adjustment supports adaptive control in healthy ageing.

## Results

### Older adults express constrained uncertainty modulation of evidence integration

During EEG and fMRI acquisition, participants performed a **M**ulti-**A**ttribute **A**ttention **T**ask ("MAAT"[30]; Fig. 1a, S1−0). Participants had to sample dynamic visual stimuli that varied along four features: color

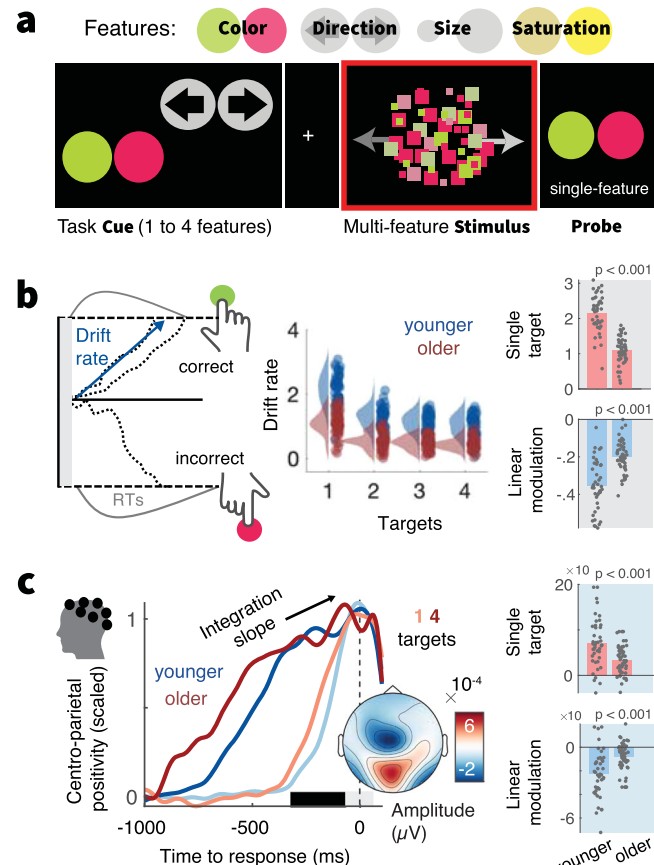

**Fig. 1 | Older adults show constrained decision-related adjustments to rising uncertainty. a** A Multi-Attribute Attention Task ("MAAT") requires participants to sample up to four visual features of a compound stimulus for a subsequent perceptual decision. On each trial, participants were first cued to the set of possible probe features (here: motion direction and color). The compound stimulus (which always included all four features) was then presented for 3 s, followed by a single-feature probe (here: prevalence of red vs. green color in the preceding stimulus). Uncertainty was manipulated as the number of target features (one to four) in the pre-stimulus cue (see also Supplementary Fig. 1). **b** Behavioral data were modeled with a drift diffusion model, in which evidence for options is accumulated with a 'drift rate'. Older adults exhibited reduced drift rates for single targets (top) and were marked by more limited drift reductions under elevated uncertainty (bottom). Data points represent individual averages across EEG and fMRI sessions. Table S1 reports within-group statistics. **c** The Centro-parietal positivity (CPP) provides an a priori neural signature of evidence accumulation. The rate of evidence accumulation was estimated as the linear slope of the CPP during the time window indicated by the black bar. Older adults exhibited reduced integration slopes for single targes (top) and were marked by constrained load-related slope shallowing under elevated uncertainty (bottom). To illustrate age- (blue: younger, red: older) and condition-differences (color saturation) in integration slope, responses have been rescaled to the [0, 1] range for visualization. Supplementary Fig. 6 shows original traces. p-values result from two-sided independent t-tests (see Statistical analyses). YA: $N = 42$. OA: $N = 53$.

(green/red), movement direction (left/right), size (small/large), and color saturation (low/high). Stimuli were presented for three seconds, after which participants had to indicate the more prevalent of two options for a single probed feature. Valid pre-stimulus cues indicated which features could be probed on the current trial. Uncertainty was parametrically manipulated by increasing the number of cued features[50,51]. When participants received a single cue, they could attend to only a single target feature during stimulus presentation (low uncertainty); whereas multi-feature cues reduced information about which feature would be probed, thus necessitating (extra-dimensional) attention switches[52,53] between up to four target features ("target load";

high uncertainty) to optimally inform probe-related decisions. Younger and older adults performed above chance level for all visual features (Supplementary Fig. 2).

To characterize probe-related decision processes, we fitted a hierarchical drift-diffusion model[54] (HDDM) to participants' responses. The model estimates (a) the drift rate at which evidence is integrated towards a decision bound, (b) the distance between correct and incorrect decision bounds, and (a) the non-decision time of probe processing and response execution. Across sessions and age groups the best fitting models consistently included uncertainty effects in all three parameters (Supplementary Fig. 3). Here, we focused on the drift rate based on its close association to sampled evidence[30]. Supplementary Note 1 reports the remaining parameters. With rising uncertainty, drift rates decreased for both age groups, indicating that uncertainty generally constrained choice evidence for the probed feature. Crucially, relative to younger adults, older participants' drift rates were reduced when only a single feature was cued as relevant and decreased less alongside increasing uncertainty (Fig. 1b). These effects remained present when only features with age-matched single-target accuracies were included in the model (Supplementary Note 2). In relative terms, such dampened adjustment reflected larger relative performance decreases when transitioning into more uncertain contexts in older than younger adults (Supplementary Note 3). Neither accuracy nor drift rate variations between individual features could account for the observed age effects (Supplementary Note 4).

We assessed the convergence of behavioral results with an a priori neural proxy signature of evidence integration, the slope of the EEG's centroparietal positive potential (CPP[55]; Fig. 1c, Supplementary Fig. 6) prior to decisions. Consistent with behavioral modeling, CPP slopes were flatter for older relative to younger participants in single-target contexts, and older adults' uncertainty-related modulation of CPP slopes was minimal (Fig. 1c). In line with both indices capturing latent evidence integration, CPP and drift estimates were inter-individually related, both for single targets (t(93) = 5.72, $p < 0.001$, $r = 0.51$, 95%CI = [0.34,0.64]; *age-partial:* t(92) = 3.49, $p < 0.001$, $r = 0.34$, 95%CI = [0.14,0.5]), and their uncertainty modulation (t(93) = 4.86, $p < 0.001$, $r = 0.45$ 95%CI = [0.27,0.59]; *age-partial:* t(92) = 2.70, $p = 0.008$, $r = 0.27$, 95%CI = [0.08,0.45]; Supplementary Fig. 6). We also investigated contralateral beta power as a signature of motor response preparation[56] but did not observe clear relations to drift rate or CPP estimates (Supplementary Note 5), suggesting that it may be a less suitable evidence integration index here. Reduced modulation of pre-response slopes in older adults was observed (at both central and parietal sites) also after controlling for overlapping potentials locked to probe onset (Supplementary Note 6). Taken together, older adults' decisions were marked by reduced evidence integration rates for single targets, and more constrained absolute drift rate reductions under uncertainty.

## Decoding indicates uncertainty-induced trade-offs between feature specificity and sensitivity

Higher single-target drift rates and larger drift reductions may reflect an adaptive trade-off between reduced single-target specificity and elevated sensitivity to *multiple* features under higher uncertainty. However, as decisions were linked to the probed feature, they cannot elucidate how unprobed features were processed. To clarify this question, we performed fMRI decoding analyses. We created pairwise classifiers that targeted the sensory representation of each feature's prevalent option (e.g., left vs. rightward movement) based on BOLD responses in visual cortex (see *Methods: fMRI decoding of prevalent feature options*). The prevalent option of individual features could be decoded above chance during stimulus presentation (Fig. 2a). Robust decoding was observed for all cued features except for saturation, for which discrimination was also behaviorally most challenging (Supplementary Fig. 2). In line with task-relevance motivating feature representations[18], above-chance decoding for uncued features was not

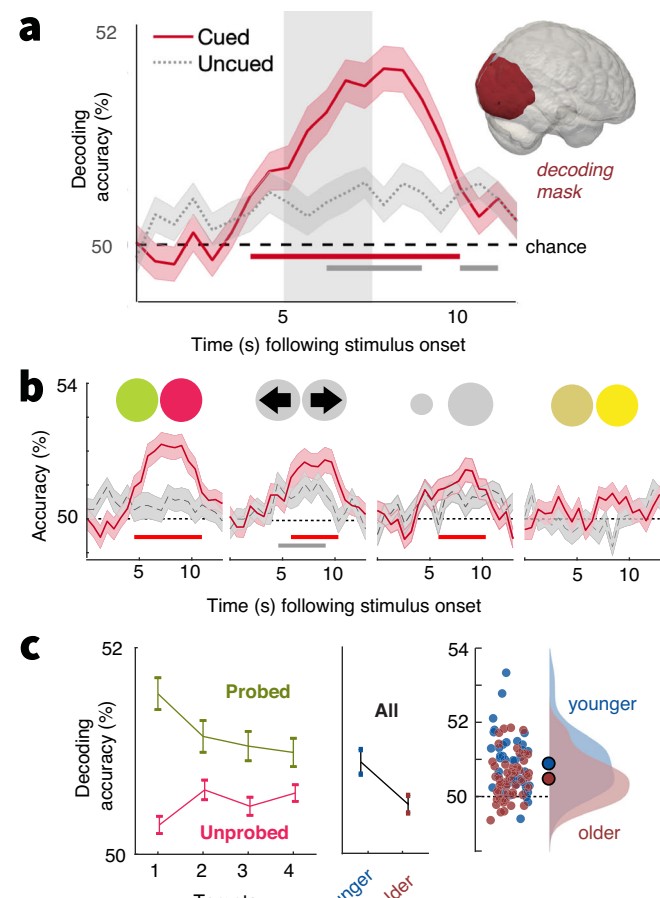

**Fig. 2 | Decoding of prevalent options from visual cortex. a** Decoding accuracy for cued and uncued features across age groups (means ± SEM; *N* = 93). Gray shading indicates the approximate timing of stimulus presentation considering the temporal lag in the hemodynamic response. Lines indicate periods of statistically significant differences from chance decoding accuracy (50%) as assessed by cluster-based permutation tests. The inset highlights the visual cortex mask from which signals were extracted for decoding. **b** Same as in a, but for separate feature probes. Bars indicate sign. above-chance accuracy during the approximate time of stimulus presentation. **c** Decoding accuracy for probed and unprobed features as a function of the number of cued targets; and decoding accuracy for all features as a function of age. Accuracy was averaged across significant decoding timepoints for cued features. Means ± within-subject SEM for (un)probed features, means ± SEM for age analysis (younger *N* = 42, older *N* = 51). Plots illustrate in-text statistical results derived from linear mixed effects models (see methods: fMRI decoding of prevalent feature options).

statistically significant above-chance in the same time window of interest, except for motion discrimination (see Fig. 2b).

Next, we assessed uncertainty and age effects on decoding accuracy. First, we applied classifiers to trials in which target features were probed, which mirrors the behavioral task. A linear mixed effects model indicated a significant reduction in decoding accuracy with increasing uncertainty (t(17762) = −3.56, $\beta = −0.18$, SE = 0.05, $p < 0.001$; Fig. 2c), as well as reduced decoding accuracy for older adults (t(91) = −2.77, $\beta = −0.862$, SE = 0.31, $p = 0.007$). No statistically significant interaction was observed (t(17761) = −0.31, $\beta = −0.03$, SE = 0.1, $p = 0.760$). Crucially, such uncertainty-related precision losses may trade-off against sensitivity to other cued, but ultimately unprobed features. We tested this possibility by considering decoding accuracy across all *unprobed* features in any given trial. This analysis indicated that uncertainty indeed slightly increased decoding accuracy across unprobed features (t(17762) = 2.94, $\beta = 0.077$, SE = 0.026, $p = 0.003$). Decoding accuracy trended to be lower in older compared to younger

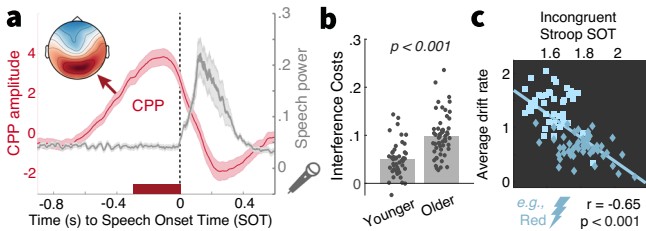

**Fig. 3 | MAAT evidence integration relates to prepotent response inhibition. a** Centro-Parietal Positivity (CPP) traces and speech signal power suggest high validity for the semi-automatically labeled speech onset times (SOTs). The CPP trace has been averaged across age and congruency conditions and displays means ± SEM (*N* = 98). The inset shows the mean EEG topography during the final 300 ms prior to speech onset. **b** The voiced Stroop task indicated robust interference costs whose magnitude was larger in older adults. Table S1 reports within-group statistics. **c** Participants with larger MAAT drift rates showed faster responses to incongruent trials (e.g., responding blue to the inset stimulus; two-sided Pearson correlation), also after accounting for categorical age (squares: younger; diamonds: older adults) and covariation with congruent SOTs (see main text).

adults (t(91) = −1.92, $\beta$ = −0.259, SE = 0.134, *p* = 0.057). Again, no statistically significant interaction was observed (t(17761) = 0.78, $\beta$ = 0.04, SE = 0.05, *p* = 0.434). Consistent with a mixture of opposing uncertainty effects on probed and unprobed features, no statistically significant uncertainty effect was observed when all trials were considered (t(17762) = 0.53, $\beta$ = 0.012, SE = 0.024, *p* = 0.593), but decoding accuracy was globally reduced in older adults (t(91) = −2.84, $\beta$ = −0.41, SE = 0.144, *p* = 0.006). Decoding analyses thus suggest that rising uncertainty increased sensitivity to more diverse features in both age groups, albeit at the cost of reduced precision for single features.

## MAAT performance generalizes to feature selection in the context of low perceptual demands

Relative to younger adults, older adults appear to have encoded less single-target evidence for downstream decisions. However, the multifaceted demands of the MAAT do not resolve whether such differences arise from task idiosyncrasies such as the necessity to resolve high perceptual uncertainty for each feature, or whether they capture differences related to flexible feature selection. To adjudicate between these accounts, participants also performed a Stroop task, which probes the capacity to inhibit prepotent responses to one of two features (the color vs. semantics) of a presented word[57]. We recorded voice responses as a more naturalistic modality for older adults[58]. To estimate speech onset times (SOTs ~ reaction times), we labeled the voice onset in each trial's recording (see methods). Labeled SOTs showed high validity as the neural CPP peaked immediately prior to SOTs (Fig. 3a). In line with the Stroop literature[58], older adults incurred larger behavioral interference costs (Fig. 3b) than younger adults. These behavioral results were mirrored by neural CPP slopes: interference made pre-response CPP slopes shallower in both age groups, but to a larger extent in older adults, and the magnitude of individual slope reductions tracked behavioral interference costs (Supplementary Fig. 9). Crucially, participants with higher MAAT drift rates were also faster responders in the incongruent condition (Fig. 3c), pointing to a better capacity to inhibit prepotent responses. Notably, relations between MAAT drift rates and SOTs in the Stroop interference condition (t(93) = −8.25, *p* < 0.001, *r* = −0.65, 95%CI = [−0.75,−0.51]) held after controlling for age and SOTs in the congruent condition (t(91) = −2.92, *p* = 0.005, *r* = −0.29, 95%CI = [−0.46,−0.09]). The opposite relation between congruent SOTs and drift rates was not statistically robust after accounting for age and incongruent SOTs (congruent SOTs-drift: t(93) = −4.21, *p* < 0.001, *r* = −0.4, 95%CI = [−0.56,−0.22], age- and incongruent SOT-partial: t(91) = 1.26, *p* = 0.202, *r* = 0.13, 95%CI =

[−0.07,0.33]). As such, selective inhibition of interfering features, as opposed to processing speed, appears to be a key contributor to individual MAAT drift rates. Taken together, these findings suggest that individual and age differences in MAAT drift rates generalize to flexible feature selection also in perceptually unambiguous contexts.

## Theta power and pupil diameter upregulation with elevated uncertainty dampens in old age

Our results indicate age-related constraints in perceptual and decision adjustment to uncertainty. To test whether such constraints are rooted in a reduced neural uncertainty response as expected under a maintenance account of cognitive and brain aging, we assessed several a priori signatures (see ref. 30) during MAAT stimulus presentation by means of two-group task partial-least-squares analyses (PLS, see methods). First, we assessed the effect of uncertainty on frontocentral theta power, an index of cognitive control[59] and exploration under uncertainty[60]. Uncertainty increased theta power in both age groups (Fig. 4a), but to a lesser extent in older adults (Fig. 4a). Next, we assessed phasic changes in pupil diameter, a signature that covaries with neuromodulation and arousal[61,62], has been related to frontal control[2,29,30,63,64], and is sensitive to rising demands[65] such as dynamically changing and uncertain contexts[28,66]. Once again, we observed that uncertainty increased pupil diameter in both age groups, with more constrained upregulation in older adults (Fig. 4b). This effect could not be explained by a "spill-over" of differential luminance responses during the cueing phase (Supplementary Note 7). The magnitude of pupil modulation was related to individual theta power increases (t(98) = 2.89, *p* = 0.005, *r* = 0.28, 95%CI = [0.09, 0.45]; age-partial: t(97) = 1.92, *p* = 0.042, *r* = 0.19, 95%CI = [0, 0.38]), indicating a joint uncertainty modulation. These results indicate that both age groups were sensitive to rising uncertainty, albeit older adults to a dampened extent.

## Only younger adults adjust posterior cortical excitability to varying uncertainty

Elevated uncertainty may impact perception by altering sensory excitability. To test this, we focused on three indices related to cortical excitability: alpha power, sample entropy, and aperiodic 1/f slopes[30,67]. We constrained analyses to posterior sensors as we targeted visual-parietal cortices. Supplementary Note 8 reports whole-channel analyses. In younger adults, we observed uncertainty effects on all three signatures (Fig. 5a–c), akin to those we previously reported[30]. In line with putative excitability increases, posterior alpha power decreased alongside uncertainty, while sample entropy increased and the aperiodic spectral slope shallowed. However, we found no evidence of a similar modulation in older adults for any of the probed signatures (Fig. 5, Supplementary Fig. 10), indicating a failure of the aged system to adjust to changing uncertainty demands. Such failure may be rooted in a less precise estimation of environmental uncertainty in the aged neural system[16]. However, we reduced inference demands in our design by providing overt cues on each trial, and keeping the cue set identical for eight consecutive trials. In line with age-invariant sensitivity to uncertainty cues, we observed comparable increases in pre-stimulus alpha power alongside uncertainty in both age groups (Supplementary Note 9). However, these increases were not associated with subsequent behavioral drift rate adjustments, arguing against a direct role of pre-stimulus alpha power in adjudicating uncertainty. We additionally considered the steady-state visual evoked potential (SSVEP) as a proxy of bottom-up processing. Despite robust and comparable SSVEPs in both age groups, we found no evidence of uncertainty modulation in either group (Supplementary Note 10). Given that the 30 Hz flicker frequency was shared between all stimulus features, this suggests that sensory processing of the compound stimulus was similar between uncertainty conditions and age groups. Taken together, our results suggest that older adults may have suffered from a relative failure to adjust perceptual excitability to

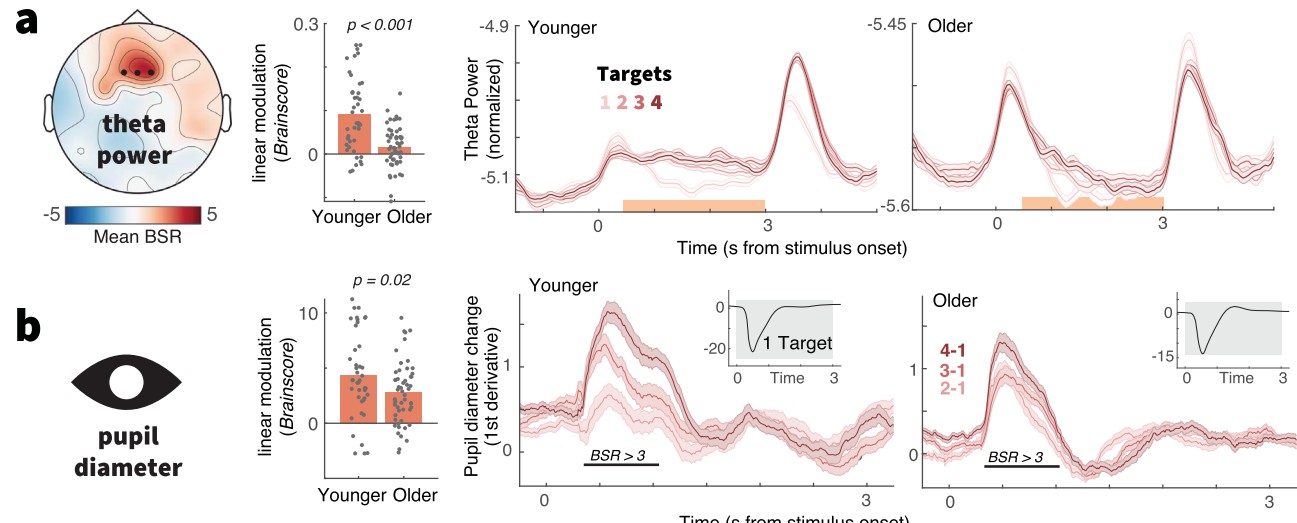

**Fig. 4 | EEG and pupil markers of control demands.** Uncertainty increases theta power (**a**) and pupil diameter (**b**) across the adult lifespan, but increases are attenuated in older age. (Left) The topography indicates mean bootstrap ratios (BSR) from the task partial least squares (PLS) model. "*Brainscores*" summarize the expression of this pattern into a single score for each condition and participant (see methods; Supplementary Fig. 10 shows condition-wise *Brainscores*). (Center) Age comparison of linear *Brainscore* changes under uncertainty (~age x load interaction; *p*-values refer to unpaired t-tests). Both signatures exhibited significant uncertainty modulation in younger, as well as older adults (as assessed via one-sample t-tests; see Table S1), with constrained modulation in older adults. (Right) Time series data are presented as means ± within-subject S.E.Ms. Target amount corresponds to increasing color saturation. Orange shading in a indicates the timepoints across which data have been averaged for the task PLS. Black lines in (**b**) indicate time points exceeding a BSR of 3 (-99% threshold). The uncertainty modulation of pupil diameter occurred on top of a general pupil constriction due to stimulus-evoked changes in luminance upon task onset (see inset). Luminance did by stimulus design not systematically differ across load levels.

changing feature relevance, rather than insensitivity to uncertainty information or an inability to encode the undifferentiated stimulus.

## BOLD modulation links neuro-behavioral responses to uncertainty across the adult lifespan

Finally, we investigated uncertainty-related changes in whole-brain fMRI BOLD activation during stimulus presentation, extending sensitivity also to subcortical areas like the thalamus that are considered critical for managing task uncertainty[30,68,69]. We targeted associations between uncertainty-related BOLD modulation and the a priori neurobehavioral signatures (i.e., uncertainty-induced changes in drift rate, theta power, pupil diameter, alpha power, 1/f slopes, and sample entropy) using a multivariate behavioral PLS analysis (see Methods; Supplementary Note 11 reports a task PLS targeting the main effect of uncertainty). We identified a single latent variable (LV; *permuted* < 0.001) with positive frontoparietal and thalamic loadings, and most pronounced negative loadings in medial PFC and hippocampus (Fig. 6a, Table S5). Loadings on this inter-individual difference LV resembled the main effect of uncertainty on BOLD activation (Supplementary Fig. 15). Older adults expressed this LV to a lesser extent than younger adults as indicated by lower fMRI *Brainscores* (Fig. 6b), indicating dampened BOLD modulation in the face of changing uncertainty. *Brainscores* were associated with the latent score of neurobehavioral input signatures (Fig. 6c), such that less dampened BOLD modulation in older age tracked a larger modulation of decision, EEG, and pupil signatures. Figure 6d depicts relations to the individual signatures of the model: across age groups, greater BOLD modulation corresponded to larger drift rate reductions, more pronounced theta power and pupil diameter increases, and larger excitability modulation (see Supplementary Fig. 16 for more signatures). Brainscores did not significantly vary by gender (Supplementary Fig. 16b). As the PLS model captured variance both within and across age groups, we used linear-mixed-effects models to assess the age-dependency of these relations. These models indicated that all a priori signatures, except sample entropy and 1/f modulation, predicted *Brainscores* also after

accounting for the shared main effects of age (Table 1). This indicates a robust coupling of uncertainty effects between most signatures, while aligning with unobserved posterior excitability modulation in older adults. Control analyses indicate that within- and between-group differences in BOLD modulation did not reflect differential choice difficulty (i.e., accuracy) for individual features (Supplementary Fig. 17).

Behavioral relations were closely tracked by thalamic BOLD activation. To obtain insights within this differentiated structure, we assessed regional loadings based on projection zones and nucleus segmentations (Fig. 6e). Loadings were highest in subregions with frontoparietal projections, including the mediodorsal nucleus (Fig. 6f). In contrast, a traditional visual "relay" nucleus of the thalamus, the lateral geniculate nucleus, did not show sensitivity to our uncertainty manipulation (Fig. 6f). This indicates a specificity of thalamic effects that coheres with functional subdivisions and alludes to uncertainty-invariant sensory processing of the compound stimulus. These results indicate that the mediodorsal thalamus contributes to maintained uncertainty adjustments across the adult lifespan.

Task uncertainty is a contextual challenge[17] that necessitates flexible control, including attentional and working memory adjustment. We probed whether the fMRI activation observed here can be reduced to either of these processes. In line with our operationalization capturing latent uncertainty, reverse inference analyses indicate relations between spatial loadings of the behavioral PLS and prior "state entropy"[29] activation and meta-analytic "uncertainty" maps. This overlap was larger than with either "working memory" or "attention" maps (Supplementary Note 12), suggesting that task uncertainty introduces multifaceted demands[70] that do not fully converge with traditional working memory or attention manipulations (Supplementary Note 13).

## Discussion

Managing uncertainty is vital for navigating the flux of life. While some environments help us to prioritize specific inputs over others, many contexts provide few, contrasting, or ambiguous cues. Here, we

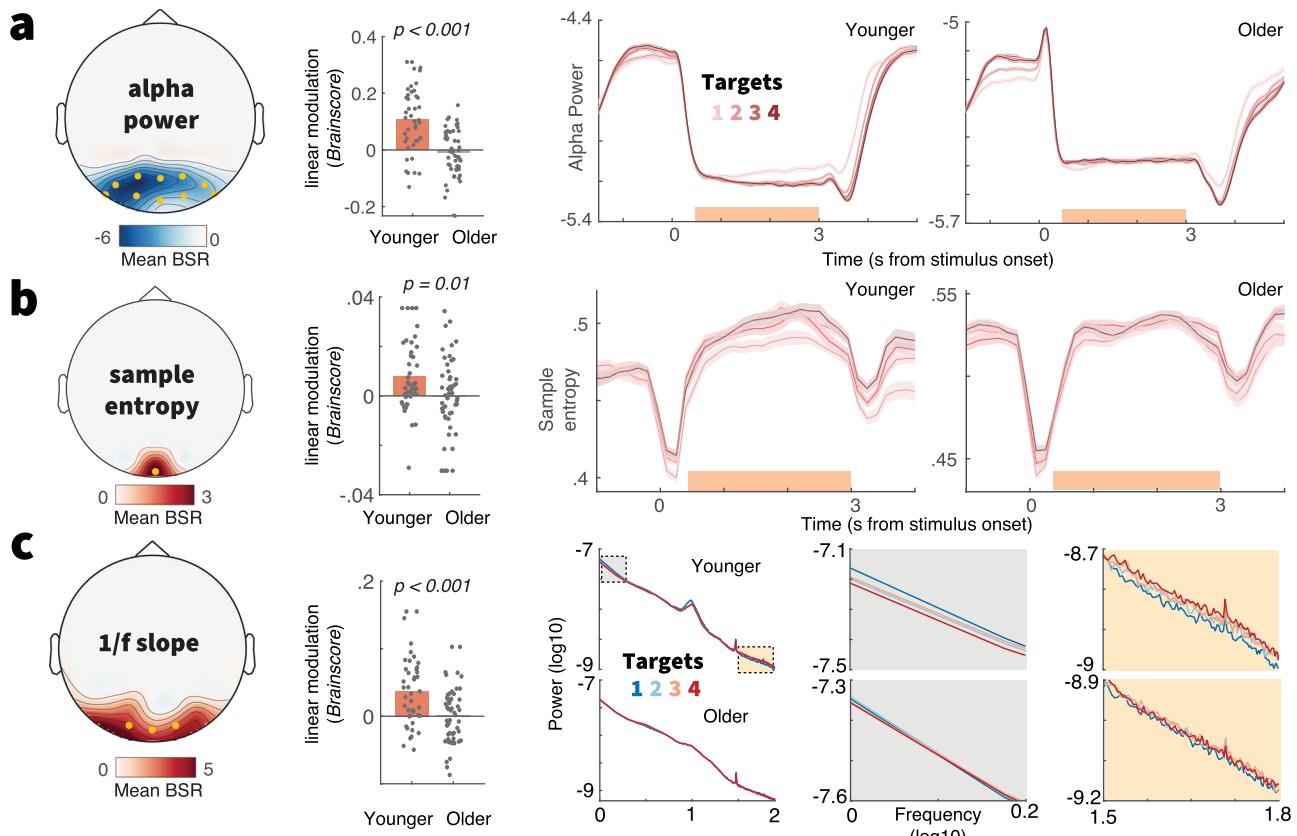

**Fig. 5 | Only younger adults upregulate cortical excitability under increased uncertainty.** Results of task partial least squares (PLS) models, assessing relations of alpha power (**a**), sample entropy (**b**) and aperiodic 1/f slope (**c**) to uncertainty. (Left) Topographies indicate mean bootstrap ratios (BSR). Orange dots indicate the sensors across which data were averaged for data visualization. (Center) Age comparison of linear uncertainty effects (-age x uncertainty interaction). Statistics refer to two-sided unpaired t-tests (younger $N = 42$, older $N = 53$; see Statistical analyses). For condition-wise *Brainscores*, see Supplementary Fig. 10. All three signatures exhibited significant uncertainty modulation in younger, but not in older adults. Table S1 reports within-group statistics. (Right) Time series data for a and b are presented as means ± within-subject S.E.M.s (younger $N = 47$, older $N = 53$). Orange shading indicates the timepoints across which data have been averaged for the respective task-PLS. For (**c**), average spectra during stimulus presentation are shown as a function of the number of targets. Plots with gray and orange background highlight low- and high-frequency ranges, respectively.

manipulate task uncertainty via unambiguous cues that are repeated on each trial. This design allows us to ask how task uncertainty impacts downstream processing, in contrast with prior designs that ask how perceptually ambiguous task cues impact processing of unambiguous inputs[68,71–73]. We show that healthy older adults exhibit markedly dampened adaptations to explicit uncertainty variations across coupled EEG/fMRI/pupil signatures. Our results thereby extend observations that older adults rely less on uncertainty representations to guide internal computations[16] by characterizing several plausible mechanisms for this shortfall. Specifically, our results suggest that such computational constraints do not exclusively stem from an inadequate sensitivity to latent uncertainty, as overt uncertainty cues were similarly processed across age groups. Rather, our findings support the maintenance account of cognitive and brain aging[74], wherein individuals with brain responses more closely resembling younger adults also more dynamically adjust perceptual and decision computations according to momentary uncertainty.

**Age differences in selecting features of multi-task stimuli**
In our retro-cue design, evidence integration towards perceptual choices indirectly indexes how multi-task stimuli were processed. Older adults showed reduced modulation of evidence integration as a function of uncertainty but were also marked by reduced drift rates in response to single-target cues. This is consistent with age-related problems of goal selection in the context of inherently ambiguous

multi-task stimuli[13,14,75]. Mayr (2001) indicated that "even when people have complete knowledge about the type of action to perform in the immediate future, they have problems implementing this knowledge in an optimal manner when more than one action rule may be relevant in principle" (p. 105). The MAAT's multi-dimensional stimuli constantly feature such rule ambiguity, thus requiring internal segregation and prioritization among possible task goals. A question concerns the relation of such "global set-selection costs" to working memory capacity[13,14,75], given that multi-task stimuli (and their cues) also require maintenance of larger task sets. While the MAAT does not fully resolve this debate (Supplementary Note 13), it uses single-trial cues and homogeneous cue blocks to limit working memory demands. As such, results for the single-target condition conceptually replicate prior observations of large age differences in static set selection costs. In tandem, our uncertainty manipulation indicates age differences in *dynamic* task set management, indicated by reduced adjustment of downstream decision processes and larger relative performance costs in older as compared to younger adults.

**Fronto-thalamic circuits may enable stable and flexible feature selection across the adult lifespan**
As part of the neural uncertainty response, we observed a behaviorally relevant upregulation of anterior cingulate cortex (ACC) BOLD activation and (presumably ACC-based[59,76]) mediofrontal theta power. By charting the progression through multiple task contexts[77–79], the ACC

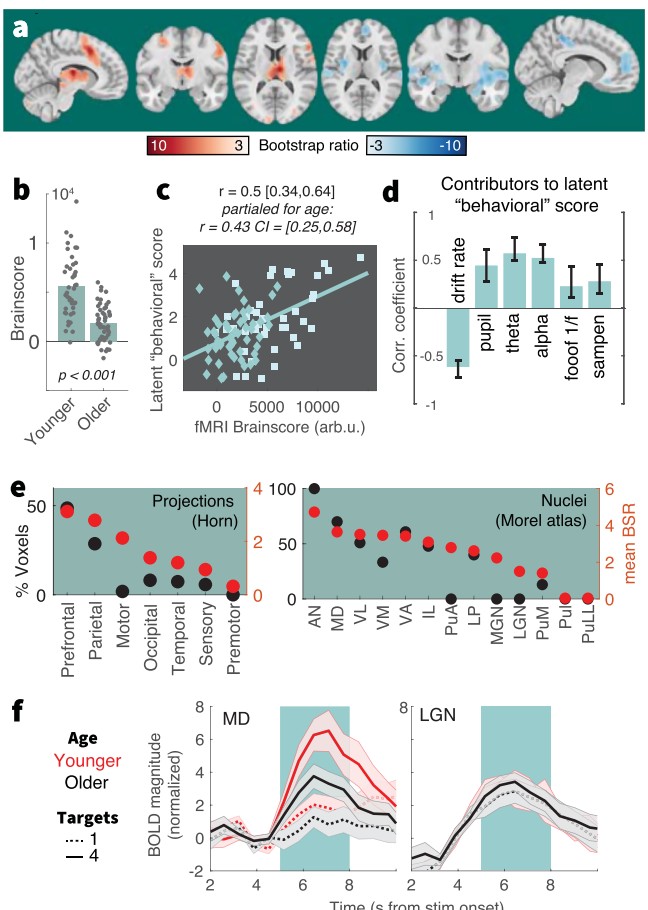

**Fig. 6 | Multivariate relation of EEG/pupil/behavioral signatures to fMRI BOLD uncertainty modulation. a** Results of a behavioral partial least squares (PLS) analysis linking linear changes in BOLD activation to interindividual EEG, pupil, and behavioral differences. Table S4 lists peak coordinates. **b** The multivariate expression of BOLD changes alongside rising uncertainty was reduced in older ($N = 53$) compared with younger adults ($N = 42$). Table S1 reports within-group statistics. **c** Individual *fMRI Brainscore* differences related to behavioral composite scores, also after accounting for age covariation. Squares = younger individuals; diamonds = older individuals. **d** Contributing signatures to the *fMRI Brainscore*. All signature estimates refer to linear uncertainty changes. Data are presented as mean values ± bootstrapped 95% confidence intervals ($N = 1000$ bootstraps). **e** Major nuclei and projection zones in which behavioral relations are maximally reliable according to average Bootstrap ratios (red) and the percentage of voxels in each subregion exceeding a BSR of 3. See Methods for abbreviations. Strongest expression is observed in nuclei that project to fronto-parietal cortical targets. **f** Visualization of uncertainty modulation for the mediodorsal nucleus, a "higher order" nucleus, and the LGN, a visual relay nucleus. Traces display mean ± SEM, for younger (red) and older adults (black), and varying target amount (broken: single, continuous: four). The green shading indicates the approximate stimulus presentation period after accounting for the delay in the hemodynamic response function.

**Table 1 | Summary of Brainscore predictors, while controlling for categorical age**

| Predictor | t-value | p value | partial η² | 95% CI |
|---|---|---|---|---|
| Behavioral score | 4.6043 | <0.001 | 0.1962 | [468, 1178] |
| age | −6.3809 | <0.001 | 0.3192 | [−4023,−2113] |
| Drift mod. | −4.3334 | <0.001 | 0.2308 | [−13515, −5020] |
| age | −3.9624 | <0.001 | 0.2006 | [−3405, −1131] |
| Pupil mod. | 4.171 | <0.001 | 0.1622 | [155, 437] |
| age | −6.7664 | <0.001 | 0.3375 | [−4206, −2297] |
| Theta mod. | 4.2533 | <0.001 | 0.2005 | [7852, 21,609] |
| age | −4.8662 | <0.001 | 0.2471 | [−3664, −1540] |
| Alpha mod. | 3.2185 | 0.002 | 0.1294 | [3055, 12,900] |
| age | −4.934 | <0.001 | 0.2589 | [−3901, −1662] |
| 1/f mod. | 0.10914 | 0.91 | 1.4e-04 | [−10,338, 11,540] |
| age | −6.7591 | <0.001 | 0.3574 | [−4782, −2610] |
| SampEn mod. | 1.5944 | 0.11 | 0.0279 | [−6618, 60,491] |
| age | −6.7385 | <0.001 | 0.3390 | [−4534, −2470] |

Separate linear-mixed-effects models assessed effects of target signature, categorical age, and age x signature interactions on Brainscores. We observed no significant interaction in any of the models (all $p > 0.05$), pointing to consistent relations across age groups; therefore, all reported models only include main effects of signature and age. Supplementary Fig. 16 reports similar results using partial regressions. Degrees of freedom: 92 (all models).

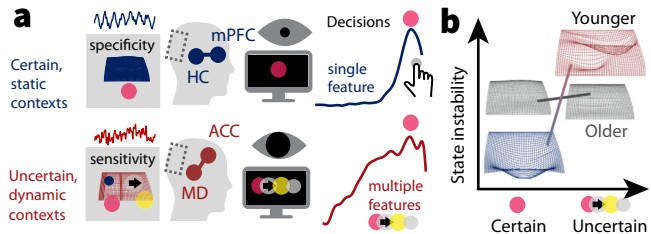

**Fig. 7 | Schematic model summary. a** In static contexts, prefrontal-hippocampal networks may signal high confidence in the current task state, which enables stable task sets, and a targeted processing of specific sensory representations with high acuity. Such selective processing of specific task-relevant features benefits their efficient evidence integration. Such selectivity would be suboptimal in contexts with uncertain or changing task sets, however. An MD-ACC circuit may track such uncertainty and enhance stochastic task set flexibility in changing or ambiguous contexts. In coordination with posterior-parietal cortex, this feasibly enables more diverse albeit less precise perceptual representations. **b** The neural system of younger adults may more dynamically adjust feature fidelity during stimulus presentation to the degree of uncertainty. Observed effects align with a switch between a specific high-acuity processing of individual features (blue), and a more diverse, if less precise processing of multiple features (red; see also Thiele & Bellgrove, 2018). In contrast, the aged neural system may be stuck in a suboptimal middle ground that affords neither stable precision, nor flexible imprecision. mPFC medial prefrontal cortex, HC hippocampus, ACC anterior cingulate cortex, MD mediodorsal thalamus.

can estimate contextual volatility[80] and uncertainty[16,81] to guide exploration of alternative goals, strategies, and attentional targets[60,82–84]. Non-human animal studies suggest that high task uncertainty switches ACC dynamics to a state of increased excitability[67,85] and stochastic activity[86], which benefits concurrent sensitivity to alternate task rules[87]. Also in humans, the ACC is sensitive to stimulus features before they behaviorally guide task strategies[86,88], suggesting that the ACC contributes to the exploration of alternate task strategies[89,90]. While our results align with such contribution, we also localize high uncertainty sensitivity in the mediodorsal (MD) thalamus, which aligns with the MD being a key partner for selecting,

switching, and maintaining cortical task representations[23,91,92] especially in uncertain contexts that require multifaceted computational adjustments[30,68,69]. Extrapolating from this emerging perspective, the MD-ACC circuit may regulate the extent of task set stability vs. flexibility[93–95] according to contextual demands (Fig. 7a). Partial evidence for such a notion is provided by models that link task stability in low-uncertainty contexts to thalamic engagement[96]. The current observations suggest a complementary thalamic role in flexible task set management. While maintained across the adult lifespan, BOLD and theta power signals indicated that such MD-ACC upregulation dampened in older age[97,98]. Indeed, the ACC network is particularly

susceptible to age-related metabolic declines[35–37] as well as structural atrophy[44]. Retained ACC function on the other hand is a hallmark of cognitive reserve[99], relates to maintained executive function[37], and is a fruitful target of cognitive interventions in older adults[98]. Given evidence of a key role of the MD thalamus in the coordination of ACC engagement and our observations of reduced MD-ACC sensitivity to uncertainty in older age, the thalamus may be an underappreciated site for cascading age-related deficits in cognitive stability and flexibility.

## Neuromodulation may sculpt the dynamic range of uncertainty adjustments

Neurotransmitter systems provide a candidate substrate for computational adjustments under uncertainty. In response to rising uncertainty, phasic norepinephrine release can sensitize the system to incoming signals[100,101] by increasing neuro-behavioral activation[61,102,103]. Pupil diameter, an index that is partially sensitive to noradrenergic drive[65], robustly increases alongside uncertainty during learning[28] and attention[104], environmental exploration[105], and change points in dynamic environments[28,66,106]. Notably, increases have been observed in contexts that require an agent to learn more or less about a single option[107]; i.e., conditions in which sensitivity for *one* option increases. Here, pupil increases precede *decreases* in evidence integration for single features. Under the notion that uncertainty requires exploration of a larger space of options, we argue that this is akin to a lower learning rate for an individual feature at the benefit of distributed learning across uncertain features. Non-selective gain increases, e.g., provided by global arousal, can favor such distributed learning[108]. We observe that pupil sensitivity to rising uncertainty is retained across the adult lifespan but dampens in older age. Such dampening hints at declining noradrenergic responsiveness in older age[41,109,110], arising from reduced LC integrity[111,112], and/or decreased LC engagement[113]. Notably, pupil sensitivity to volatility has been related to the ACC as a primary source of cortical LC input[27,114], and joint increases of ACC activation and pupil diameter in uncertain, or dynamic contexts has consistently been observed in studies that record both signals[2,29,30,63,64]. While future studies need to clarify the origin of constrained pupil adjustments in older age, our results affirm the relevance of the extended LC system for attentional function across the lifespan[41]. In contrast to noradrenaline's potential role in sensitizing, cholinergic innervation from the basal forebrain may foster selectivity via cortical gain increases[115,116]. Notably, basal forebrain BOLD activation decreased under uncertainty alongside regions such as the medial prefrontal cortex and hippocampus, that are sensitive to subjective confidence[117], suggesting that these regions may support stable task beliefs when uncertainty is low[85,118,119] (Fig. 7a). The constrained BOLD modulation observed in older adults may thus point to reduced task set stability in low-uncertainty contexts (Fig. 7b)[11], plausibly because of limited cholinergic gain control. Similar ideas have been captured in the cortical gain theory of aging[120], but in the context of the dopamine system[39,121]. Computational models and pharmacological studies indeed support a role of dopamine availability in task set stability and flexibility[122,123]. For instance, amphetamines (operating via the DA system) can in- and decrease task set stability in ACC[124,125] depending on baseline dopamine levels in frontoparietal cortex and thalamus[126]. Given that our results align with the fronto-thalamic system being a primary neural substrate of cognitive aging[39,45,127], the potential contribution of age-related dopamine depletion to constrained uncertainty adjustments deserves future clarification.

## Excitability as a neural mechanism for acuity/sensitivity trade-offs

Uncertainty motivates sensitivity to multiple features at the cost of selective precision (or "acuity")[3]. Our decoding results cohere with this notion, suggesting that representational fidelity depends on whether a feature is included in the current task set[18], but also on competition with other elements for shared neuro-computational resources[128]. Excitability changes in parietal/sensory cortices provide a candidate neural implementation for such trade-off. One index of (decreased) cortical excitability is alpha power. Models suggest that broad alpha power increases reflect active inhibition of irrelevant information[129–133], while targeted alpha desynchronization can selectively disinhibit relevant information[44]. With advancing adult age, alpha power decreases broadly, which has been linked to inhibitory filtering deficits[41,134–137] that manifest in maladaptive sensitivity also to irrelevant[7] and non-salient features[138] of compound stimuli[6]. Decoding and decision analyses indeed indicate that older adults' task performance suffered from reduced single-target information, in line with selective filtering deficits[139,140]. Alpha desynchronization, in turn, is thought to reflect increased sensitivity when multiple input features[26] must be jointly tracked[141,142] and retained in working memory[143–146]. In addition to alpha power, aperiodic dynamics such as the spectral slope of the EEG potential[147] and signal entropy[148] may also index levels of neural excitability[67,147]. Here, we reproduce uncertainty-guided excitability increases as indexed by all three signatures in younger adults[30], but find no evidence for a comparable modulation in older age. Such deficit may be rooted in age-related declines of GABAergic inhibition[42,43]. Aperiodic dynamics at rest suggest increased excitatory tone in older age[149–151], including in the current sample[148]. Such imbalances[152] may constrain the dynamic range of excitability modulation in older age, both on- and off-task[33,153]. It is also possible that the consistently high level of perceptual uncertainty, i.e., the difficulty of arbitrating between choice options of each feature, was overly taxing especially for older participants. Based on behavioral and decoding results, younger adults were indeed better able to arbitrate feature-specific options across uncertainty levels, relative to older adults. In this scenario, preserved excitability modulation may be observed if choice evidence was less ambiguous for individual features. However, performance on the Stroop task suggests that age-related deficits (and individual differences) in feature selection generalize to contexts of low perceptual ambiguity. Moreover, variations in perceptual difficulty across features could not explain inter-individual and age differences in neural uncertainty modulation. As perceptual uncertainty resolution relies on partially dissociable circuits from those implicated in feature selection[154–156], future work needs to chart the ability to resolve either type across the lifespan.

## The role of working memory

It is notoriously challenging to distinguish the explanatory power of competing functional mechanisms that could explain age-related differences in cognition. In the current paradigm the manipulation of uncertainty was accomplished by varying the number of potentially relevant features, which arguably may also increase working memory load. However, there are several reasons why we believe that uncertainty is the primary driver of the observed pattern of results. First, the increase of age differences was greatest when transitioning from one to two possible features. While both one and two features should remain well within working memory capacity, the difference between these two conditions is highly significant on the uncertainty dimension (i.e., the contrast between certainty and uncertainty). Further, our reversed-inference analyses indicate that the neuroanatomical results are more consistent with age effects in uncertainty processing than in working-memory functioning. On a more theoretical level, it is important to note that when it comes to aging, working memory is not a simple, unidimensional construct. For example, the fact that age-independent individual differences and age differences express themselves in markedly different manners[157–159], makes this construct much less attractive as a general, candidate mechanism. Instead, an age-related failure to dynamically respond to varying uncertainty has the potential of providing a unifying explanation of age differences

across paradigms and domains, and thus can serve as a useful lens on healthy cognitive ageing. Given that uncertainty provides a signal for adaptive control per se, our observation that uncertainty-based control provides a principled challenge for the aged brain highlights the need to better understand how the brain estimates and computationally leverages uncertainty signals across the lifespan.

## Methods

### Sample

47 healthy young adults (mean age = 25.8 years, SD = 4.6, range 18 to 35 years; 25 women) and 53 healthy older adults (mean age = 68.7 years, SD = 4.2, range 59 to 78 years; 28 women) performed a perceptual decision task during 64-channel active scalp EEG acquisition. 42 younger adults and all older adults returned for a subsequent 3 T fMRI session. We recruited a combined total of $N = 100$ participants, with approximately age-matched and gender-matched sample sizes informed by our prior inter-individual work[30]. Gender of participants was determined based on self-report. Participants were recruited from the participant database of the Max Planck Institute for Human Development, Berlin, Germany (MPIB). Participants were right-handed, as assessed with a modified version of the Edinburgh Handedness Inventory[160], and had normal or corrected-to-normal vision. Participants reported to be in good health with no known history of neurological or psychiatric incidences and were paid for their participation (10 € per hour). All older adults had Mini Mental State Examination (MMSE)[161,162] scores above 25. The ethics board of the Deutsche Gesellschaft für Psychologie (DGPs) approved the study protocol ("Flexibles Denken - State Switch"). All participants gave their written informed consent prior to participating in the study.

### Procedure: EEG Session

Participants were seated 60 cm in front of a monitor in an acoustically and electrically shielded chamber with their heads placed on a chin rest. Following electrode placement, participants were instructed to rest with their eyes open and closed, each for 3 min. Afterwards, participants performed a Stroop task (see below), followed by the visual attention task instruction & practice (see below), the performance of the task and a second Stroop assessment. Stimuli were presented on a 60 Hz 1920 × 1080 p LCD screen (AG Neovo X24) using PsychToolbox 3.0.11[163–165]. The session lasted ~3 h. EEG was continuously recorded from 60 active (Ag/AgCl) electrodes using BrainAmp amplifiers (Brain Products GmbH, Gilching, Germany). Scalp electrodes were arranged within an elastic cap (EASYCAP GmbH, Herrsching, Germany) according to the 10% system[166], with the ground placed at AFz. To monitor eye movements, two additional electrodes were placed on the outer canthi (horizontal EOG) and one electrode below the left eye (vertical EOG). During recording, all electrodes were referenced to the right mastoid electrode, while the left mastoid electrode was recorded as an additional channel. Online, signals were digitized at a sampling rate of 1 kHz. In addition to EEG, we simultaneously tracked eye movements and assessed pupil diameter using EyeLink 1000+ hardware (SR Research, v.4.594) with a sampling rate of 1 kHz.

### Procedure: MRI session

A second testing session included structural and functional MRI assessments. First, participants received a short refresh of the task ("MAAT", see below) instructions and practiced the task outside the scanner. Then, participants were placed in the TimTrio 3 T scanner and were instructed in the button mapping. We collected the following sequences: T1w, task (4 runs), T2w, resting state, DTI, with a 15 min out-of-scanner break following the task acquisition. The session lasted ~3 h. Whole-brain task fMRI data (4 runs á ~11.5 min, 1066 volumes per run) were collected via a 3 T Siemens TrioTim MRI system (Erlangen, Germany) using a multi-band EPI sequence (factor 4; TR = 645 ms; TE = 30 ms; flip angle 60°; FoV = 222 mm; voxel size 3 × 3× 3 mm; 40

transverse slices. The first 12 volumes (12 × 645 ms = 7.7 s) were removed to ensure a steady state of tissue magnetization (total remaining volumes = 1054 per run). A T1-weighted structural scan (MPRAGE: TR = 2500 ms; TE = 4.77 ms; flip angle 7°; FoV = 256 mm; voxel size 1 × 1 × 1 mm; 192 sagittal slices) and a T2-weighted structural scan were also acquired (GRAPPA: TR = 3200 ms; TE = 347 ms; FoV = 256 mm; voxel size 1 × 1 × 1 mm; 176 sagittal slices).

### The multi-attribute attention task ("MAAT")

The MAAT requires participants to sample up to four visual features in a compound stimulus, in the absence of systematic variation in bottom-up visual stimulation (see Fig. 1). Participants were shown a dynamic stimulus that combined four features of visual squares: their color (red/green), movement direction (left, right), size (small, large) and saturation (low, high). The task incorporates features from random dot motion tasks which have been extensively studied in both animal models[167–169] and humans[55,170]. Following stimulus presentation, a probe queried the prevalence of one feature (e.g., color: whether the stimulus contained more red or green squares) via 2-AFC (alternative forced choice). Before stimulus onset, a valid cue informed participants about the feature set, out of which the probe feature would be selected. We parametrically manipulated task uncertainty by including between one and four features in the cue. Participants were instructed to respond as fast and accurately as possible to increase their chance of bonus. They were instructed to use cue information to guide their attention during stimulus presentation between "focusing on a single feature" vs. "considering multiple features" to optimally prepare for the upcoming probe.

The perceptual difficulty of each feature was determined by (a) sensory differences between the two options and (b) the relative evidence for either option. For (a) the following values were used: high (RGB: 128, 255, 0) and low saturation green (RGB: 192, 255, 128) and high (RGB: 255, 0, 43) and low saturated red (RGB: 255, 128, 149) for color and saturation, 5 and 8 pixels for size differences and a coherence of 0.2 for directions. For (b) the relative choice evidence was chosen as follows: color: 60/40; direction: 80/20; size: 65/35; saturation: 60/40. Parameter difficulty was established in a pilot population, with the aim to produce above-chance accuracy for individual features. Parameters were held constant across age groups to equate bottom-up inputs.

Each session included four approx. 10 min task runs, each including eight blocks of eight trials (i.e., a total of 32 trial blocks; 256 trials). The size and constellation of the cue set was held constant within eight-trial blocks to reduce set switching and working memory demands. At the onset of each block, the valid cue set, composed of one to four target features, was presented for 5 s. Each trial was structured as follows: recuing phase (1 s), fixation phase (2 s), dynamic stimulus phase (3 s), probe phase (incl. response; 2 s); ITI (un-jittered; 1.5 s). At the offset of each block, participants received performance feedback for 3 s. The four features spanned a constellation of 16 feature combinations (4 × 4), of which presentation frequency was matched within participants. The size and type of the cue set was pseudo-randomized: Within each run, every set size was presented once, but never directly following a block of the same set size. In every block, each feature in the active set acted as a probe in at least one trial. Moreover, any feature served as a probe equally often across blocks. The dominant options for each feature were counterbalanced across all trials of the experiment. To retain high motivation during the task and encourage fast and accurate responses, we instructed participants that one response would randomly be drawn at the end of each block; if this response was correct and faster than the mean RT during the preceding block, they would earn a reward of 20 cents. However, we pseudo-randomized feedback such that all participants received an additional fixed payout of 10 € per session. This bonus was paid at the end of the second session, at which point participants were debriefed.

## Stroop performance

Participants performed a voiced Stroop task before and after the main MAAT task in the EEG session. EEG signals were acquired during task performance. Two older adults did not complete the second Stroop acquisition. In the Stroop task, we presented three words (RED, GREEN, BLUE) either in the congruent or incongruent display color. Each of the two runs consisted of 81 trials, with fully matched combinations, i.e., 1/3rd congruent trials. Stimuli were presented for two seconds, followed by a one-second ITI with a centrally presented fixation cross. Participants were instructed to indicate the displayed color as fast and accurately as possible following stimulus onset by speaking into a microphone. During analysis, speech on- and offsets were pre-labeled automatically using a custom tool (**C**omputer-**A**ssisted **R**esponse **L**abeler (CARL); doi: 10.5281/zenodo.7505622), and manually inspected and refined by one of two trained labelers. Voiced responses were manually labeled using the CARL GUI. Speech onset times (SOTs) were highly reliable across two Stroop sessions preceding and following the MAAT ($r = 0.83$, $p = 5e\text{-}26$), as were individual interference costs ($r = 0.64$, $p = 5e\text{-}13$). We therefore averaged SOTs estimates across both runs, where available. For EEG analyses, single-trial time series were aligned to SOTs, and averaged according to coherence conditions. The centroparietal positive potential was extracted from channel POz, at which we observed a maximum potential during the average 300 ms prior to SOT (see inset in Fig. 3a).

## Behavioral estimates of probe-related decision processes

Sequential sampling models, such as the drift-diffusion model, have been used to characterize evolving perceptual decisions in 2-AFC random dot motion tasks[55], memory retrieval[171], and probabilistic decision making[172]. We estimated individual evidence integration parameters within the HDDM 0.6.0 toolbox[54] to regularize relatively sparse within-subject data with group priors based on a large number of participants. Premature responses faster than 250 ms were excluded prior to modeling, and the probability of outliers was set to 5%. 7000 Markov-Chain Monte Carlo samples were sampled to estimate parameters, with the first 5000 samples being discarded as burn-in to achieve convergence. We judged convergence for each model by visually assessing both Markov chain convergence and posterior predictive fits. Individual estimates were averaged across the remaining 2000 samples for follow-up analyses. We fitted data to correct and incorrect RTs (termed "accuracy coding" in Wiecki, et al.[54]). To explain differences in decision components, we compared four separate models. In the 'full model', we allowed the following parameters to vary between conditions: (i) the mean drift rate across trials, (ii) the threshold separation between the two decision bounds, (iii) the non-decision time, which represents the summed duration of sensory encoding and response execution. In the remaining models, we reduced model complexity, by only varying (a) drift, (b) drift + threshold, or (c) drift + NDT, with a null model fixing all three parameters. For model comparison, we first used the Deviance Information Criterion (DIC) to select the model which provided the best fit to our data. The DIC compares models based on the maximal log-likelihood value, while penalizing model complexity. The full model provided the best fit to the empirical data based on the DIC index (Supplementary Fig. 3c) in both the EEG and the fMRI session, and in either age group. Posterior predictive checks indicated a suitable recovery of behavioral effects using this full solution. Given the observation of high reliability between sessions[30] (see Supplementary Fig. 3), we averaged parameter estimates across the EEG and fMRI sessions for the main analysis. In contrast with previous work[30], we did not constrain boundary separation estimates[173] here given our observation of CPP threshold differences in older adults. See also Supplementary Note 1 for a brief discussion of NDT and boundary separation.

## EEG preprocessing

Preprocessing and analysis of EEG data were conducted with the FieldTrip toolbox (v.20170904)[174] and using custom-written MATLAB (The MathWorks Inc., Natick, MA, USA) code. Offline, EEG data were filtered using a 4th order Butterworth filter with a passband of 0.5 to 100 Hz. Subsequently, data were downsampled to 500 Hz and all channels were re-referenced to mathematically averaged mastoids. Blink, movement and heart-beat artifacts were identified using Independent Component Analysis (ICA)[175]; and removed from the signal. Artifact-contaminated channels (determined across epochs) were automatically detected using (a) the FASTER algorithm[176], and by (b) detecting outliers exceeding three standard deviations of the kurtosis of the distribution of power values in each epoch within low (0.2–2 Hz) or high (30–100 Hz) frequency bands, respectively. Rejected channels were interpolated using spherical splines[177]. Subsequently, noisy epochs were likewise excluded based on a custom implementation of FASTER and on recursive outlier detection. Finally, recordings were segmented to stimulus onsets and were epoched into separate trials. To enhance spatial specificity, scalp current density estimates were derived via 4th order spherical splines[177] using a standard 1005 channel layout (conductivity: 0.33 S/m; regularization: 1^-05; 14th degree polynomials).

## Electrophysiological estimates of probe-related decision processes

**Centro-Parietal Positivity (CPP).** The Centro-Parietal Positivity (CPP) is an electrophysiological signature of internal evidence-to-bound accumulation[55,173,178]. We investigated the task modulation of this established signature and assessed its convergence with behavioral parameter estimates. To derive the CPP, preprocessed EEG data were low-pass filtered at 8 Hz with a 6th order Butterworth filter to exclude low-frequency oscillations, epoched relative to response and averaged across trials within each condition. In accordance with the literature, this revealed a dipolar scalp potential that exhibited a positive peak over parietal channel POz (Fig. 1c). We temporally normalized individual CPP estimates to a condition-specific baseline during the final 250 ms preceding probe onset. As a proxy of evidence drift rate, CPP slopes were estimates via linear regression from −250 ms to −100 ms surrounding response execution, while the average CPP amplitude from −50 ms to 50 ms served as an indicator of decision thresholds (i.e., boundary separation; e.g.,[173]).

**Contralateral mu-beta.** Decreases in contralateral mu-beta power provide a complementary, effector-specific signature of evidence integration[56,173]. We estimated mu-beta power using 7-cycle wavelets for the 8–25 Hz range with a step size of 50 ms. Spectral power was time-locked to probe presentation and response execution. We re-mapped channels to describe data recorded contra- and ipsi-lateral to the executed motor response in each trial, and averaged data from those channels to derive grand average mu-beta time courses. Individual average mu-beta time series were baseline-corrected using the −400 to −200 ms prior to probe onset, separately for each condition. For contralateral motor responses, remapped sites C3/5 and CP3/CP5 were selected based on the grand average topography for lateralized response executions (see inset in Supplementary Fig. 7a). Mu-beta slopes were estimated via linear regression from −250 ms to −50 ms prior to response execution, while the average power from −50 ms to 50 ms indexed decision thresholds (e.g.,[173]).

## Electrophysiological indices of top-down modulation during sensation

**Low-frequency alpha and theta power.** We estimated low-frequency power via a 7-cycle wavelet transform (linearly spaced center frequencies; 1 Hz steps; 2 to 15 Hz). The step size of estimates was 50 ms, ranging from −1.5 s prior to cue onset to 3.5 s following stimulus offset.

Estimates were log10-transformed at the single trial level[179], with no explicit baseline-correction. For statistics, data were averaged across time windows of interest (see respective Figure captions) and entered a task-PLS analysis (see "Multivariate partial least squares analyses") to quantify the magnitude of power modulation as a function of target load without the need to pre-specify relevant channels.

**Steady State Visual Evoked Potential (SSVEP).** The SSVEP characterizes the phase-locked, entrained visual activity (here 30 Hz) during dynamic stimulus updates (e.g.,[180]). These features differentiate it from induced broadband activity or muscle artefacts in similar frequency bands. We used these properties to normalize individual single-trial SSVEP responses prior to averaging: (a) we calculated an FFT for overlapping one second epochs with a step size of 100 ms (Hanning-based multitaper) and averaged them within each uncertainty condition; (b) spectrally normalized 30 Hz estimates by subtracting the average of estimates at 28 and 32 Hz, effectively removing broadband effects (i.e., aperiodic slopes), and; (c) we subtracted a temporal baseline −700 to −100 ms prior to stimulus onset. Linear uncertainty effects on SSVEPs were assessed by paired t-tests on linear uncertainty slope estimates across posterior channel averages.

**Time-resolved sample entropy.** Sample entropy[181] quantifies the irregularity of a time series of length $N$ by assessing the conditional probability that two sequences of $m$ consecutive data points will remain similar when another sample ($m + 1$) is included in the sequence (for a visual example see Fig. 1a in ref. [148]). Sample entropy is defined as the inverse natural logarithm of this conditional similarity: The similarity criterion ($r$) defines the tolerance within which two points are considered similar and is defined relative to the standard deviation (-variance) of the signal (here set to $r = 0.5$). We set the sequence length $m$ to 2, in line with previous applications[148]. An adapted version of sample entropy calculations implemented in the mMSE toolbox (available from https://github.com/LNDG/mMSE; 10.5281/ zenodo.4672138) was used[148,182], wherein entropy is estimated across discontinuous data segments to provide time-resolved estimates. The estimation of scale-wise entropy across trials allows for an estimation of coarse scale entropy also for short time-bins (i.e., without requiring long, continuous signals), while quickly converging with entropy estimates from continuous recordings[183]. To remove the influence of posterior-occipital low-frequency rhythms on entropy estimates, we notch-filtered the 8–15 Hz alpha band using 6th order Butterworth filter prior to the entropy calculation[148]. Time-resolved entropy estimates were calculated for 500 ms windows from −1 s pre-stimulus to 1.25 s post-probe with a step size of 150 ms. As entropy values are implicitly normalized by the variance in each time bin via the similarity criterion, no temporal baseline correction was applied.

**Aperiodic (1/f) slopes.** The aperiodic 1/f slope of neural recordings is closely related to the sample entropy of broadband signals[148] and has been suggested as a proxy for cortical excitation-inhibition balance[147]. Spectral estimates were computed by means of a Fast Fourier Transform (FFT) over the final 2.5 s of the presentation period (to exclude onset transients) for linearly spaced frequencies between 2 and 80 Hz (step size of 0.5 Hz; Hanning-tapered segments zero-padded to 20 s) and subsequently averaged. Spectral power was log10-transformed to render power values more normally distributed across participants. Power spectral density (PSD) slopes were estimated using the fooof toolbox (v1.0.0-dev) using default parameters[184].

**Pupil diameter.** Pupil diameter was recorded during the EEG session using EyeLink 1000 at a sampling rate of 1000 Hz and was analyzed using FieldTrip and custom-written MATLAB scripts. Blinks were automatically indicated by the EyeLink software (version 4.40). To increase the sensitivity to periods of partially occluded pupils or eye

movements, the first derivative of eye-tracker-based vertical eye movements was calculated, z-standardized, and outliers >=3 STD were removed. We additionally removed data within 150 ms preceding or following indicated outliers. Finally, missing data were linearly interpolated, and data were epoched to 3.5 s prior to stimulus onset to 1 s following stimulus offset. We quantified phasic arousal responses via the rate of change of pupil diameter traces as this measure (i) has higher temporal precision and (ii) has been more strongly associated with noradrenergic responses than the overall response[185]. We down-sampled pupil timeseries to 100 Hz. First derivative pupil traces were smoothed using a 300 ms moving median. For statistics, timeseries were entered into a task-PLS (see "Multivariate partial least squares analyses") to quantify the magnitude of pupil modulation as a function of target load without the need to pre-specify a relevant time window.

### fMRI-based analyses

**Preprocessing of functional MRI data.** fMRI data were preprocessed with FSL 5 (RRID:SCR_002823)[186,187]. Pre-processing included motion correction using McFLIRT, smoothing (7 mm) and high-pass filtering (.01 Hz) using an 8th order zero-phase Butterworth filter applied using MATLAB's filtfilt function. We registered individual functional runs to the individual, ANTs brain-extracted T2w images (6 DOF), to T1w images (6 DOF) and finally to 3 mm standard space (ICBM 2009c MNI152 nonlinear symmetric)[188] using nonlinear transformations in ANTs 2.1.0[189] (for one participant, no T2w image was acquired and 6 DOF transformation of BOLD data was preformed directly to the T1w structural scan). We then masked the functional data with the ICBM 2009c GM tissue prior (thresholded at a probability of 0.25), and detrended the functional images (up to a cubic trend) using SPM12's spm_detrend. We also used a series of extended preprocessing steps to further reduce potential non-neural artifacts[153,190]. Specifically, we examined data within-subject, within-run via spatial independent component analysis (ICA) as implemented in FSL-MELODIC[191]. Due to the high multiband data dimensionality in the absence of low-pass filtering, we constrained the solution to 30 components per participant. Noise components were identified according to several key criteria: (a) Spiking (components dominated by abrupt time series spikes); (b) Motion (prominent edge or "ringing" effects, sometimes [but not always] accompanied by large time series spikes); (c) Susceptibility and flow artifacts (prominent air-tissue boundary or sinus activation; typically represents cardio/respiratory effects); (d) White matter (WM) and ventricle activation[192]; (e) Low-frequency signal drift[193]; (f) High power in high-frequency ranges unlikely to represent neural activity ($\geq 75\%$ of total spectral power present above 0.10 Hz;); and (g) Spatial distribution ("spotty" or "speckled" spatial pattern that appears scattered randomly across $\geq 25\%$ of the brain, with few if any clusters with $\geq 80$ contiguous voxels). Examples of these various components we typically deem to be noise can be found in ref. [194]. By default, we utilized a conservative set of rejection criteria; if manual classification decisions were challenging due to mixing of "signal" and "noise" in a single component, we generally elected to keep such components. Three independent raters of noise components were utilized; >90% inter-rater reliability was required on separate data before denoising decisions were made on the current data. Components identified as artifacts were then regressed from corresponding fMRI runs using the regfilt command in FSL. To reduce the influence of motion and physiological fluctuations, we regressed FSL's 6 DOF motion parameters from the data, in addition to average signal within white matter and CSF masks. Masks were created using 95% tissue probability thresholds to create conservative masks. Data and regressors were demeaned and linearly detrended prior to multiple linear regression for each run. To further reduce the impact of potential motion outliers, we censored significant DVARS outliers during the regression as described by ref. [195]. We calculated the 'practical significance' of DVARS estimates and applied a threshold of 5[196]. The

regression-based residuals were subsequently spectrally interpolated during DVARS outliers as described in ref. 195 and[197]. BOLD analyses were restricted to participants with both EEG and MRI data available ($N = 42$ YA, $N = 53$ OA).

**fMRI decoding of prevalent feature options.** We performed a decoding analysis to analyze the extent to which participants' visual cortices contained information about the prevalent option of each feature. $N = 2$ older adults with two missing runs each were excluded from this analysis due to the more limited number of eligible trials. We trained a decoder based on BOLD signals from within a visual cortex mask that included Jülich parcellations ranging from V1 to area MT. We resliced the mask to 3 mm and created an intersection mask with the cortical gray matter mask used throughout the remaining analyses. For classification analyses, we used linear support-vector machines (SVM)[198] implemented with libsvm (www.csie.ntu.edu.tw/~cjlin/libsvm). As no separate session was recorded, we trained classifiers based on all trials (across uncertainty conditions) in which the target feature was probed, therefore necessitating but not exhaustively capturing trials on which the respective feature was also cued. By experimental design, the number of trials in which a target feature was probed was matched across uncertainty levels. We used a bootstrap classification approach in the context of leave-one-out cross-validation to derive single-trial estimates of decoding accuracy. To increase the signal-to-noise ratio for the decoders, we averaged randomly selected trials into three folds (excluding any trial used for testing) and concatenated two pseudo-trials from each condition to create the training set. Trained decoders were then applied to the left-out trial. This train-and-test procedure was randomly repeated 100 times to create bootstrapped single-trial estimates. Finally, decoding accuracy was averaged across trials based on condition assignment (e.g., whether a given feature was cued or uncued). To assess above-chance decoding accuracy in time, we used univariate cluster-based permutation analyses (CBPAs). These univariate tests were performed by means of dependent samples t-tests, and cluster-based permutation tests[199] were performed to control for multiple comparisons. Initially, a clustering algorithm formed clusters based on significant t-tests of individual data points ($p < 0.05$, two-sided; cluster entry threshold) with the spatial constraint of a cluster covering a minimum of three neighboring channels. Then, the significance of the observed cluster-level statistic (based on the summed t-values within the cluster) was assessed by comparison to the distribution of all permutation-based cluster-level statistics. The final cluster $p$ value was assessed as the proportion of 1000 Monte Carlo iterations in which the cluster-level statistic was exceeded. Cluster significance was indicated by p-values below 0.025 (two-sided cluster significance threshold). To test uncertainty and age effects, we initially fitted linear mixed effects models with random intercepts and fixed effects of uncertainty, age, and an uncertainty x age interaction. As no significant interaction was indicated for any of the models (probed: $p = 0.760$; unprobed: $p = 0.434$; all: $p = 0.625$), we removed the interaction term for the main effect estimation. We constrained analysis to timepoints for which the cluster-based permutation analysis indicated above-chance decoding for cued features (Fig. 2a; 4.5–11.5 s post-stimulus onset). We focused on probed and unprobed feature trials, as they are matched in trial number at each uncertainty level.

**BOLD uncertainty modulation and relation to multi-modal signatures.** We conducted a 1st level analysis using SPM12 to identify beta weights for each condition separately. Design variables included stimulus presentation (4 volumes; separate regressors for each uncertainty condition; parametrically modulated by sequence position), onset cue (no mod.), and probe (2 volumes, parametric modulation by RT). Design variables were convolved with a canonical HRF, including its temporal derivative as a nuisance term. Nuisance regressors included 24 motion parameters[200], as well as continuous DVARS estimates. Autoregressive modeling was implemented via FAST. Output beta images for each uncertainty condition were finally averaged across runs. At the group (2nd) level, we examined the relationship between voxel-wise 1st level beta weights and uncertainty conditions within a task PLS analysis; and probed links between linear BOLD modulation and interindividual differences in multi-modal signatures of interest via a behavioral PLS (see *Multivariate partial least squares analyses*). For visualization, spatial clusters were defined based on a minimum distance of 10 mm, and by exceeding a size of 25 voxels. We identified regions associated with peak activity based on cytoarchitectonic probabilistic maps implemented in the SPM Anatomy Toolbox (Version 2.2c)[201]. If no assignment was found, the most proximal assignment to the peak coordinates was reported.

**Temporal dynamics of thalamic engagement.** To visualize the uncertainty modulation of thalamic activity, we extracted signals within a binary mask of thalamic divisions extracted from the Morel atlas[202]. Preprocessed BOLD timeseries were segmented into trials, spanning the period from the stimulus onset to the onset of the feedback phase. Given a time-to-peak of a canonical hemodynamic response function (HRF) between 5 and 6 s, we designated the 3 s interval from 5 to 8 s following the stimulus onset trigger as the stimulus presentation interval, and the 2 s interval from 3 to 5 s as the fixation interval, respectively. Single-trial time series were then temporally normalized to the temporal average during the approximate fixation interval.

**Thalamic loci of behavioral PLS.** To assess the thalamic loci of most reliable behavioral relations, we assessed bootstrap ratios within two thalamic masks. First, for nucleic subdivisions, we used the Morel parcellation scheme as consolidated and kindly provided by Hwang et al. [203] for 3 mm data at 3 T field strength. The abbreviations are as follows: AN: anterior nucleus; VM: ventromedial; VL: ventrolateral; MGN: medial geniculate nucleus; LGN: lateral geniculate nucleus; MD: mediodorsal; PuA: anterior pulvinar; LP: lateral-posterior; IL: intralaminar; VA: ventral-anterior; PuM: medial pulvinar; Pul: pulvinar proper; PuL: lateral pulvinar. Second, to assess cortical white-matter projections we considered the overlap with seven structurally derived cortical projection zones suggested by Horn & Blankenburg[204], which were derived from a large adult sample ($N = 169$). We binarized continuous probability maps at a relative 75% threshold of the respective maximum probability, and re-sliced masks to 3 mm (ICBM 2009c MNI152).

**Statistical analyses**

**Outlier handling.** For each signature, we defined outliers at the subject-level as individuals within their respective age group whose values (e.g., estimates of linear modulation) exceeded three scaled median absolute deviations (MAD) as implemented in MATLAB. Such individual data points were winsorized prior to statistical analysis. For repeated measures analyses, such individuals were removed prior to statistical assessment.

**Linear uncertainty effect estimates.** To estimate the linear uncertainty modulation of dependent variables, we calculated 1st level beta estimates ($y = $ intercept + β*target load + e) and assessed the slope difference from zero at the within-group level (see Table S1) using two-sided paired t-tests. Similarly, we compared linear uncertainty effect estimates between groups using two-sided unpaired t-tests. We assessed the relation of individual linear load effects between measures of interest via Pearson correlations.

**Within-subject centering.** To visually emphasize effects within participants, we use within-subject centering across repeated measures

conditions by subtracting individual cross-condition means and adding global group means. For these visualizations, only the mean of the dependent values directly reflects the original units of measurement, as individual data points by construction do not reflect between-subject variation averaged across conditions. This procedure equals the creation of within-subject standard errors[205]. Within-subject centering is exclusively used for display and explicitly noted in the respective legends.

**Multivariate partial least squares analyses.** For data with a high-dimensional structure, we performed multivariate partial least squares analyses. PLS is a multivariate statistical technique used to identify relationships between two sets of variables. In neuroimaging studies, task PLS is often employed to relate brain activity (measured by techniques like fMRI, EEG, or MEG) to experimental conditions (task PLS) or behavioral measures (behavioral PLS)[206,207].

To assess main effects of uncertainty, we performed Task PLS analyses. Task PLS begins by calculating a between-subject covariance matrix (COV) between conditions and a 'neural' index. This covariance matrix is then decomposed using singular value decomposition (SVD). This yields a left singular vector of experimental condition weights (U), a right singular vector of brain weights (V), and a diagonal matrix of singular values (S). Task PLS produces orthogonal latent variables (LVs) that reflect optimal relations between experimental conditions (e.g., target load) and (neural) data of interest. We ran a task PLS version in which group means were removed from condition means to highlight how conditions were modulated by group membership, i.e., condition and condition-by-group effects. Separate task PLS analyses were performed for 'neural' values of theta power (Fig. 4), pupil diameter (Fig. 4), excitability signatures (Fig. 5), and pre-stimulus alpha power (Supplementary Fig. 14), fMRI BOLD (Supplementary Fig. 15).

To examine multivariate relations between BOLD signal changes under uncertainty and interindividual differences in decision, excitability, and pupil modulation, we performed a behavioral PLS analysis (Fig. 6). This analysis initially calculates a between-subject correlation matrix (CORR) between (1) a 'neural' index and (2) a 'behavioral' variable of interest (although called 'behavioral', this variable can reflect any variable of interest). As the neural index, we estimated linear coefficients between 1st level beta estimates ~uncertainty, fitted within each voxel. As behavioral variables, we included the signatures reported on the left of Fig. 6c, incl. drift estimates, pupil diameter, spectral power, and excitability indices). Analogous to task PLS, CORR is decomposed using SVD: $SVD_{CORR} = USV'$, which produces a matrix of left singular vectors of behavioral weights (U), a matrix of right singular vectors of neural weights (V), and a diagonal matrix of singular values (S).

Across PLS variants, each LV (ordered strongest to weakest in S) is characterized by a data pattern that depicts the strongest available relation between brain and conditions/behavioral data. Significance of detected relations of both PLS model types was assessed using 1000 permutation tests of the singular value corresponding to the LV. Subsequent bootstrapping indicated the robustness of within-LV neural saliences across 1000 data resamples[208]. By dividing each brain weight (from V) by its bootstrapped standard error, we obtained "bootstrap ratios" (BSRs) as normalized robustness estimates. We generally threshold BSRs at values of ±3.00 (~99.9% confidence interval). We obtained a summary measure of each participant- and condition-wise expression of a LV's pattern (a "Brainscore") by multiplying the vector of weights (V) by each participant's and condition's vector of input data values (P): $Brainscore = VP'$. To summarize uncertainty modulation, task PLS Brainscores were analyzed as described above ("Linear uncertainty effect estimates").

**Reporting summary**
Further information on research design is available in the Nature Portfolio Reporting Summary linked to this article.

## Data availability
The raw EEG, fMRI, and behavioral data generated in this study have been deposited as DataLad datasets (https://doi.org/10.5281/zenodo.14264868). Structural MRI data are available under restricted access for data privacy reasons as per obtained informed consent. Defaced structural MRI data can be obtained after signing an access agreement with Research Data Management (rdm@mpib-berlin.mpg.de). Interested parties should contact the corresponding authors for more information.

## Code availability
Experimental task code is available from https://doi.org/10.5281/zenodo.14216065. Analysis code is available from https://doi.org/10.5281/zenodo.14221999.

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

## Acknowledgements

This study was conducted within the 'Lifespan Neural Dynamics Group' at the Max Planck UCL Centre for Computational Psychiatry and Ageing Research in the Max Planck Institute for Human Development (MPIB) in Berlin, Germany. DDG was supported by an Emmy Noether Programme grant from the German Research Foundation. UL acknowledges financial support from the Intramural Innovation Fund of the Max Planck Society. JQK, DDG, and UL were partially supported by the Max Planck UCL Centre for Computational Psychiatry and Ageing Research. The participating institutions are the Max Planck Institute for Human Development, Berlin, Germany, and University College London, London, UK. For more information, see https://www.mps-ucl-centre.mpg.de/en/comp2psych. UM's contribution was funded through NSF grant 2120712. The funders had no role in study design, data collection and analysis, decision to publish, or preparation of the manuscript. We thank our research assistants and participants for their contributions.

## Author contributions

J.Q.K.: Conceptualization, Methodology, Investigation, Software, Formal analysis, Visualization, Writing – original draft, Writing – review and editing, Validation, Data Curation; U.M.: Conceptualization, Writing – review and editing, U.L.: Conceptualization, Resources, Writing – review and editing, Supervision, Funding acquisition; D.D.G.: Conceptualization, Methodology, Software, Resources, Writing—review and editing, Supervision, Project administration, Funding acquisition.

## Funding

## Competing interests

The authors declare no competing interests.
