## [Peer Review File · Nature Communications]

REVIEWER COMMENTS

Reviewer #1 (Remarks to the Author):

Review Summary

Thank you for inviting me to review this paper on “Broadscale dampening of uncertainty adjustment in the aging brain” by Kosciessa, Mayr, Lindenberger, and Garrett. The authors combine modeling, neuroimaging, and psychophysiology to examine evidence accumulation in a multi-attribute attention task (MAAT) that involves varying degrees of uncertainty about the currently relevant dimension. The conclusion of the paper is that older adults (OA) have more difficulties adjusting to this kind of uncertainty. In particular, their information processing is less well-tuned to single features (“specificity”; under low uncertainty) and multiple features (“sensitivity”; under high uncertainty) compared to younger adults (YA).

Overall, the paper is well-organized and clearly written. The figures illustrate the main results quite clearly.

Major

Age-matched single-target accuracies

Figure 1b shows drift rates for the single- and multi-feature conditions. It is convincing that higher drift rates for single features and strongly reduced rates for multiple features reflect a “sensitivity-specificity” trade-off. According to the analysis, YA adjust their sampling behavior more flexibly to the different levels of uncertainty.

The authors present a control analysis in the supplement (Text S1-3) to test if the results remain present when performance differences are considered. It was unclear whether the results presented in Figure S1-3 are consistent with Figure 1b. Figure S1-3 does not seem to show the “Linear mod.” analysis (blue bar graphs comparing YA and OA) but absolute and relative differences between target 2-4 and target 1. The results seem compatible, but it would be more convincing to see the same analysis in the supplementary figure. If it is the same analysis, I would recommend using the same plot styles for both analyses.

I agree with the authors that controlling for potential group differences in accuracy is essential. However, another dimension on which YA and OA should differ in the task is reaction times (RT), which also influences the DDM parameters, including the drift rate. I was wondering if results remain the same when the age-related differences in RT are also controlled. For example, the authors could match groups in a way that drift rates for single targets are similar and examine whether the uncertainty-related decrease is more substantial in YA, which would be consistent with the perspective that YA more dynamically adjust evidence accumulation. However, if groups were comparable, it might hint that such baseline differences are more important than the authors currently assume. For example, it might suggest that YA show high drift rate dynamics because they experience less perceptual uncertainty in the task. OA might show fewer dynamic adjustments because of a floor effect, as also suggested by the authors themselves.

CPP interpretation

The authors interpret the CPP findings in terms of an evidence accumulation signal. This perspective is certainly consistent with the assumption of many previous papers in perceptual decision making. However, more recently, this perspective has been called into question, and I feel like the authors should run a control analysis to ensure that the CPP truly reflects evidence accumulation. Frömer and colleagues have suggested that the CPP does not signal evidence accumulation but rather choice conflict or confidence ([link to preprint: https://www.biorxiv.org/content/10.1101/2022.08.02.502393v4](https://www.biorxiv.org/content/10.1101/2022.08.02.502393v4)). Irrespective of the exact functional interpretation, they show that the characteristic evidence accumulation signal disappears when stimulus-locked and response-locked signals are jointly accounted for. From my perspective, the authors should run a similar analysis to double-check if the evidence accumulation perspective is valid.

Decoding approach

The authors run decoding analyses using the BOLD signal, which is common practice. However, I wonder if response time is a possible confound in the analyses comparing decoding accuracies across probes and age groups (Figure 2c). For example, are decoding accuracies for one probe possibly higher because reaction times are faster than for multiple probes? Similarly, is decoding accuracy in YA higher compared to OA because their response times are faster? In these examples, longer response times might reflect greater difficulty or effort. One way to control for this possibility is to perform decoding based on beta coefficients that are extracted from a GLM that includes response times and thereby controls for difficulty or effort.

Figures 4 and 5: Brainscore

It is unclear what “Brainscore” refers to in Figures 4 and 5. Brainscore seems to be the same as “linear modulation” in Figure 1, but it is unclear. As far as I can tell, the definition of brainscore is described later (Page 9) in the context of Figure 6. The figure caption suggests that theta power and

pupil diameter are more strongly modulated by uncertainty in YA than OA, but it is unclear how this is related to the latent variable interpretation of brainscore, as described on page 9.

Minor

Integration slopes

Figure 1c illustrates CPP (or integration) slopes. In the example (left plot), it was unclear to me what feature of the curves indicates the higher slopes in YA. Taking a look into the methods clarified this, but it could be highlighted in the caption by referring to the meaning of the black bar above the x-axis.

Moreover, in the figure caption: change “slops” to “slopes” (in (c))

Text S1-3 line 75: Inconsistent quotation marks (“preference”)

Figure 2 caption: “al” instead of “all”.

Decoding results (line 147-149) in the text describing significance for uncued features: Aren’t color and size also partly significant, not only motion? At least, this seems to be implied by the darker lines.

Figure 4

BSR definition is explained in the caption of Figure 5 for the first time. So, seeing “BSR” in Figure 4 without definition is a little confusing.

Figure 4 and Discussion

The authors show pupil results, where for both YA and OA, more uncertainty is related to larger pupil size (plus the age effect). I’m not questioning this result, but I’m wondering how it relates to the literature cited in the Discussion. The authors emphasize related findings in the learning literature, such as change point detection and uncertainty-driven increases in the learning rate. These findings have in common that an agent learns more about *one* option, which is linked to larger pupil signals. For example, the agent strongly ramps up the learning rate to adjust to a change

point. However, in this work, the authors see a *decrease* in the drift rate, which may be more in line with a lower learning rate and generally more cautious behavior (which is undoubtedly adaptive in the MAAT). Especially in YA, evidence accumulation about the features is distributed (i.e., concerns multiple options), akin to a lower learning rate that distributes learning across uncertain states or options. From my perspective, the authors could consider this point in the discussion of the pupil results.

Figure S3-1 line 140: Younger “ad” change to “adults”

Table S1: Some parenthesis issues, e.g., (linear mod.)

Overall, very inspiring work!

Reviewer #2 (Remarks to the Author):

Kosciessa and colleagues present a rigorous examination of adult age differences in effects of “uncertainty adjustments” on perceptual attention and decision making. The study implements a recently developed multi-attribute perceptual task. The task manipulates the number of attributes that could be probed at the decision phase (i.e., “target load”; 1-4), with feature cues appearing prior to the stimulus presentation phase. Task performance was measured from the probe phase, in which one of the cued feature types is presented and participants respond based on which cued feature was represented to a greater degree. The behavioral analysis of decision process and examination of neural mechanisms is extremely rigorous, and leverages drift diffusion modeling, and analysis of EEG, fMRI and pupillometry data. Across measures, the findings converge on the conclusion that older adults show less correspondence between levels of target load and neural recruitment of resources to meet those task conditions. Novel findings in this study feature the role of the mediodorsal thalamus in supporting the assessed cognitive performance across adulthood. Despite these many strengths, the study is positioned as assessing uncertainty processing, but it’s not clear from my read that this construct was measured. The throughline of the experiment and critical findings is also hard to follow. My key concerns are outlined below, point-by-point.

Major comments

1. The paper needs revisions to clarify the study's contribution to the larger literature. The authors describe the study's key findings as highlighting "neural mechanisms whose maintenance plausibly enables flexible task set, perception, and decision computations across the adult lifespan." On this point, I completely agree. However, the authors situate the study in the context of variations in uncertainty processing/response/management/representation/adjustment generally. In my view, the study is more accurately framed as examining how level of (or complexity in) top-down perceptual attention impacts perceptual decisions. It seems that what is referred to as "rising uncertainty" is equivalent to varying demands – specifically, attentional demands for perceptual processing. Under this interpretation, the findings that MAAT drift rates correlate with faster responses in the Incongruent Stroop condition are completely expected. Further, the general pattern of age differences for behavior is very similar to prior studies that examined age differences in top-down cognitive control tasks – as well as the general pattern of functional neuroimaging data, including the results for older adults with "young-like" functional activation responses to task conditions (e.g., Cabeza et al., 2018. Maintenance, reserve, and compensation: the cognitive neuroscience of healthy ageing. *Nature Reviews Neuroscience*, 19(11), 701-710.) The novel element in this study is the focus on perceptual attention load and decision making, which is associated with neural mechanisms not commonly observed in prior investigations of age differences attentional/EF demands (e.g., mediodorsal thalamus). This, I find very interesting and novel in the context of cognitive aging and task demand research. In my view, this is a very strong and rigorous study that contributes novel and publishable findings, but its findings should be more directly discussed in the context of prior work on effects of cognitive task sets/demands in aging.

The (mis)classification of cognitive function investigated has further implications. Much of the existing work on "decision making under ambiguity" involves situations where individuals do not have (and never did have) access to critical information required for an optimal decision (e.g., outcome probabilities, factual knowledge). In the current study, participants are presented with all the relevant information necessary to make a correct decision. The only uncertainty is which feature will be probed, and while participants may have certainly reflected on the level of uncertainty – the nature of the task may have made it impossible to dissociate its neural mechanisms from those of task demand or access to evidence/perceptual information during integration. If the authors disagree and believe they did isolate, and measure, the psychological construct of uncertainty, I ask that they please carefully address this in a revision.

2. As I understand it, the central questions of this paper are 1) Does aging change effects feature-relevance ambiguity on goal-directed attention and decision making, and 2) If so, how? It would be most helpful if the findings were written as more direct answers to these questions. For example, it's challenging to understanding how result summaries like "Older adults express constrained

uncertainty modulation of evidence integration” can be applied to the study’s key questions. If I am misunderstanding the overall study objective, I hope that can be clarified.

3. Throughout the paper, I had some difficulty to understand exactly what was being communicated. To address this, I recommend that the language and figures should be closely reviewed and edited for clarity and simplicity.

3a. Most critically, I had difficulty in understanding the central paradigm, and what exactly participants were instructed to do.

3b. I suggest that the authors make it clear exactly how uncertainty was manipulated at the beginning of the Results narrative.

3c. I had to read the text 4 times to understand the task, and I’m still not clear about how the participants’ objective was described to them, other than they could earn a bonus for being correct and also quick in their decisions (online methods). Elaboration of task instructions (“Participants were told...”) and edits to Figure 1a and its caption in the main manuscript would help. A revision that includes more concrete examples of different trial types would also help. Using the version of the paradigm visualization from the authors’ 2021 paper would be an improvement, but naïve readers may still struggle to understand how conditions map onto uncertainty. Showing at least two trial sequences with different levels of uncertainty (target load/#cues) would be beneficial.

3d. The paper would benefit from greater consistency in the description of the critical manipulated variables (e.g., cues, features, attributes, targets, uncertainty, probe uncertainty).

3e. Early on, the reader needs to understand how exploration is operationalized and indexed in this study.

3f. Usage of complex language could be paired with real-world examples. E.g., What are examples of “contexts in which feature relevance is ambiguous and require dynamic exploration”?

3g. Specialized terms or indirect language could be replaced with more specific phrases and use of lay language. E.g., for phrases like “young-like neural recruitment/signatures/modulation” and “dampened modulation of decision processes”.

Minor comments

4. The authors should provide background to justify and set expectations for pupillometry in the context of the larger study in the Intro.

5. How did the authors determine the sample sizes?

Reviewer #3 (Remarks to the Author):

Kosciessa and colleagues examine whether aging affects behavioral and neural signatures of uncertainty adjustment using a very well-powered cross sectional study that includes behavior, EEG and fMRI. The paper builds on a previously published task and a number of neural correlates of the primary manipulation of that task. The idea is very interesting and the paper is a tour de force drawing on a lot of different methods using some innovative analysis techniques. The primary claim in this paper is that the aging brain is impaired in its ability to represent and deal effectively with increasing levels of uncertainty. While I find value in both the experiments performed and the claim being tested, I am not yet convinced that the data really support the claim. I have articulated my primary concerns below, and would be happy if the authors could address them through a revision.

1) Most of the claims about uncertainty here are based on sensitivity of behavior to cues in the task that indicate which (1,2,3,or 4) features could be probed at the end of the trial. This manipulation does, as authors suggest, affect uncertainty about the relevant feature dimension during information processing, but it also changes the effective working memory load of the decision – to be successful on the 4 dimension condition I would need to hold 4 feature values in working memory simultaneously and then select out the appropriate one in response to the probe. Thus, it feels like any age-differences in how behavior or EEG signals scale with uncertainty could also be interpreted in terms of differential working memory capacity, which is well established to decrease with aging.

2) I'm not sure that the behavioral modeling really makes a tight case for uncertainty differences either (even leaving aside the point above). The groups differ in their performance in the single feature condition – making it hard to say what one should expect for the higher load conditions. The model is fit flexibly to get different parameters for each condition – and does not incorporate the notion of uncertainty to make testable predictions about how performance would change across target loads. The interpretation of the model parameters is also muddled by the fact that participants were forced to view stimulus for 3 seconds, and thus had no need to employ a procedure for stopping evidence accumulation. Furthermore, the model fit predicts that error trials will have the same RT distribution as correct trials – which differentiates it somewhat from the memory idea posed in point 1, since failure to remember would almost certainly lead to longer RT errors. However, this aspect of the model does not seem to match the data very well... the posterior predictives in figure S1-2b suggest that participants have longer RTs than the model predicts on error trials and this is most evident for the 4 condition in older adults, precisely when you would expect memory capacity limitations to play the largest role.

3) The strongest behavioral evidence that the authors present toward their claim is in figure S1-3a, and while I find it much more compelling than the modeling results, I worry that there is a selection bias at play – in the face of noise in the accuracy measurements for each condition (number of targets), if we select based on one of those conditions, shouldn't we expect to see a much larger shift in the age effect for that condition? Can the authors verify that this procedure works if you use half data for selection of feature and other half for statistical testing, for example?

4) The fMRI decoding result is interesting, but it would be nice to see whether this result actually matches the behavior. This could be done descriptively by fitting choice with a logistic regression model that contains values of relevant and irrelevant features in order to predict choice – it could also be done in the HDDM modeling framework that the authors setup by changing the bounds from reflecting (correct/incorrect) to the actual choice options and modeling the drift rate contributions of the various sources of evidence present on each trial. As a minor point it would be useful if fMRI decoding plots were on same axis for older/younger adults.

5) Pupil data shows a very strong trend consistent with stronger conditional modulation in young adults. However, if I understand the task correctly, there is a luminance difference in the conditions that occurs 2 seconds before stimulus onset – is it possible that some of the observed changes are carryover effects from this luminance difference?

6) Figure 1C: I found it hard to understand the link between the averages on the left and the per-subject quantifications on the right, since the difference between 1 and 4 in the older adult averaged data looks just as extreme as the one for younger adults – but the quantification suggests a massive difference.

7) The whole brain analysis linking behavioral signatures of uncertainty reduction to specific patterns of uncertainty-related brain activation seems like one of the most interesting parts of this paper, yet I found the actual procedure a bit difficult to follow given that the different parts of the analysis were described in different sections of the methods and very little intuition was provided for the motivations of the exact procedures in the results. Beyond this basic lack of clarity, I was a bit concerned that the brain activations pulled out by this complex analysis might simply just be the pattern of brain activations to the manipulation of interest – rather than saying something more specific.

8) In the schematic/summary figure, the link to the attractor network changes described feel a bit tenuous to me. The attractors depicted are point attractors, whereas the behavioral data are all interpreted through DDM models, which have internal dynamics consistent with a line attractor. The point attractor idea also seems very similar to proposals about prefrontal working memory representations, which again makes me wonder how different the effects reported here are from things that have been reported with respect to age-related changes in working memory capacity and its neural correlates.

Response to reviewers

We thank the reviewers for their thoughtful comments. Based on the feedback, we revised our manuscript to increase its accessibility and performed additional control analyses when allowed by our data. We partially report results of control analyses in direct response to reviewers, under the assumption that comments and responses will be made publicly available alongside the manuscript.

Reviewer #1 (Remarks to the Author):

Thank you for inviting me to review this paper on "Broadscale dampening of uncertainty adjustment in the aging brain" by Kosciessa, Mayr, Lindenberger, and Garrett. The authors combine modeling, neuroimaging, and psychophysiology to examine evidence accumulation in a multi-attribute attention task (MAAT) that involves varying degrees of uncertainty about the currently relevant dimension. The conclusion of the paper is that older adults (OA) have more difficulties adjusting to this kind of uncertainty. In particular, their information processing is less well-tuned to single features ("specificity"; under low uncertainty) and multiple features ("sensitivity"; under high uncertainty) compared to younger adults (YA).

Overall, the paper is well-organized and clearly written. The figures illustrate the main results quite clearly.

We thank the reviewer for these encouraging comments.

Major

1. Age-matched single-target accuracies

Figure 1b shows drift rates for the single- and multi-feature conditions. It is convincing that higher drift rates for single features and strongly reduced rates for multiple features reflect a "sensitivity-specificity" trade-off. According to the analysis, YA adjust their sampling behavior more flexibly to the different levels of uncertainty.

The authors present a control analysis in the supplement (Text S1-3) to test if the results remain present when performance differences are considered. It was unclear whether the results presented in Figure S1-3 are consistent with Figure 1b. Figure S1-3 does not seem to show the "Linear mod." analysis (blue bar graphs comparing YA and OA) but absolute and relative differences between target 2-4 and target 1. The results seem compatible, but it would be more convincing to see the same analysis in the supplementary figure. If it is the same analysis, I would recommend using the same plot styles for both analyses.

Thank you for highlighting this discrepancy. Figure S1-3 now consistently shows linear modulation estimates across panels to enable direct comparisons. The results support the notion that older adults show more constrained absolute changes (which is the main estimate that is inter-individually related to neural modulation in our prior work, and here), while in relative terms showing larger performance decreases relative to their reduced baseline.

I agree with the authors that controlling for potential group differences in accuracy is essential. However, another dimension on which YA and OA should differ in the task is reaction times (RT), which also influences the DDM parameters, including the drift rate. I was wondering if results remain the same when the age-related differences in RT are also controlled. For example, the authors could match groups in a way that drift rates for single targets are similar and examine whether the uncertainty-related decrease is more substantial in YA, which would be consistent with the perspective that YA more dynamically adjust evidence accumulation. However, if groups were comparable, it might hint that such baseline differences are more important than the authors currently assume. For example, it might suggest that YA show high drift rate dynamics because they experience less perceptual uncertainty in the task. OA might show fewer dynamic adjustments because of a floor effect, as also suggested by the authors themselves.

We agree that baseline differences cannot be exhaustively resolved and are a limitation of assessing task uncertainty at high levels of perceptual uncertainty in the current design. It may also not be straightforward: If younger adults experience less perceptual uncertainty for some features than for others, they may also need to resolve more perceptual competition by inhibiting alternative features (also in the single-target condition). Ultimately, the joint, age-differential, modulation of signatures at the heart of this work occurs under the noted limitation of high perceptual uncertainty that may be perceived differently across individuals and age groups.

We have performed additional analyses that aim to better characterize performance (accuracy, rt) and drift rate as a function of which feature has been probed (as features vary in their accuracy ~ perceptual uncertainty in the design).

As suggested, we performed an additional analysis in which we attempted to match features based on HDDM single-trial drift rates (Text S1-4). In this analysis, we estimate all canonical HDDM parameters also for variations in the feature that has been probed. We note that the large parameter space may lead to some convergence issues, which cannot be thoroughly checked (which is why the model in the manuscript does not estimate parameters for separate features).

Fig. S1-4. Exploration of feature-specific accuracy and drift rate variation. (a) Single-target accuracy and drift rate for features that are sorted by decreasing single-target accuracy and drift rate. Yellow dots indicate the conditions selected for matching in b. (b) Matching based on single-target accuracies and drift rates indicates stronger *relative* performance decreases with uncertainty in older compared with adults. (c) A median split of fMRI Brainscores (trichotomized split | full lines: largest BOLD mod.; dashed lines: smallest BOLD mod.; cf. Fig. S6-3) indicates that larger neural modulation is inter-individually and group-wise linked to higher single-trial drift rates (but not differential accuracy; see also Fig. S6-3), independent of feature preference. This suggests that neural modulation is not primarily constrained due to varying choice difficulty for individual features, but rather by a global operation (e.g., cue-guided selectivity, distractor suppression, etc.) that is shared across features. Means \pm SEMs.

Like accuracy, matching feature preference such that single-target drift rates are matching indicates *larger* uncertainty-induced drift rate decreases (Fig. S1-4b). However, it is debatable to what extent this is a *reasonable* comparison: an analysis that matches “selective” performance (under low uncertainty) between age groups already removes key age differences, which in our data are also capturing neural variation. Namely, we observed a closer link between within-group fMRI modulation and drift rates, but not accuracy (compare Fig. S1-4c left and center, see also S6-3). This was independent of which feature was probed, and thus unrelated to varying perceptual uncertainty between features. This reinforces our assumption that perceptual uncertainty per se (which in psychophysics is sometimes titrated to comparable *accuracy* levels) is not an exclusive predictor of the constrained neural uncertainty modulation within or between age groups.

Fig. R1-1. RTs for the matched feature data from Fig. S1-3. Data are means + SEMs.

Median RTs for the accuracy-matched conditions reported in the supplement show robust age group differences in response speed (Fig. R1-1). Matched and unmatched features do not vary in RT for single targets, but feature accuracy dissociates response times under higher uncertainty: features with higher accuracy suffer reduced response slowing under uncertainty. This speaks against substantial speed-accuracy trade-offs, in which “perceptually more difficult” features receive fast but more random responses under high uncertainty.

2. CPP interpretation

The authors interpret the CPP findings in terms of an evidence accumulation signal. This perspective is certainly consistent with the assumption of many previous papers in perceptual decision making. However, more recently, this perspective has been called into question, and I feel like the authors should run a control analysis to ensure that the CPP truly reflects evidence accumulation. Frömer and colleagues have suggested that the CPP does not signal evidence accumulation but rather choice conflict or confidence (link to preprint: <https://www.biorxiv.org/content/10.1101/2022.08.02.502393v4>). Irrespective of the exact functional interpretation, they show that the characteristic evidence accumulation signal disappears when stimulus-locked and response-locked signals are jointly accounted for. From my perspective, the authors should run a similar analysis to double-check if the evidence accumulation perspective is valid.

Thank you for raising this emerging discussion. We are aware of this preprint, and its claim that CPP ramping may reflect valuation of the probe (i.e., the set of alternative options), rather than the relative integration of evidence towards either choice over time.

While we appreciate the extent to which the authors have taken their arguments, we believe that the rationale/model that Frömer et al use to “decorrelate” valuation processes in their preprint can be questioned on several grounds:

- To dissect the temporal dynamics of appraisal- and choice- related neural activity, they regress single trial EEG activity onto Appraisal-related and Choice-related variables. Critically, they use a *value-based decision-making task* (with substantially longer RTs as compared to perceptual decisions), and partially misidentify the presumed CPP topography (which in their data appears during what they refer to as an “appraisal” stage; whereas choice in their data relates to topographies more consistent with alpha and theta rhythm (dis-)engagement, not a “CPP” signature as they argue). While not discussed, their data would be in line with two separate perceptual evidence integration processes for each alternative item, to then allow their value comparison. In short, the relevance of their observations for perceptual decisions as made here remains ambiguous to us.
- In the domain of perceptual decisions, the authors also highlight that deconvolving 2s stimulus- (here: probe-) related activity from the time interval prior to responses eliminates pre-response ramping (i.e., the CPP). This is believable but is also expected given that differences in the speed of evidence integrations emerge quickly after probe onset, and propagate through to the eventual response, with a clear dependence on reaction time (see e.g., Figs 2 & 4 in Churchland et al., 2008, Nat. Neuro.). As such, an evidence accumulation perspective can remain valid even if response-locked ramping can be explained by stimulus-locked potentials. Whether a decorrelation approach is appropriate therefore remains debatable to us. We note that drift rate estimates (incl. CPP slope-based) in our study are more closely related to

accuracy differences rather than RTs, the latter of which Frömer et al. regard as a confounding factor for CPPs.

We cannot repeat an appraisal-related analysis here, given that we did not acquire single-trial estimates of participant's global valuation of choice sets (and our design relates to the perceptual, not value-based, decision making that the CPP has been commonly indexed for). However, there are multiple reasons why we consider such appraisal to be an insufficient explanation of our current results:

- EEG-based estimates of the integration process (e.g., CPP slopes) are correlated with HDDM-based estimates of evidence integration (Figure S1-4c) – the methodological concerns that Frömer et al. raise do not apply to HDDM estimates.
- We observe similar “ramping” dynamics for signatures in the motor system (Text S1-5). It is notable that contralateral beta power by design reflects relative information for one motor response over the other, thus fulfilling the definition of a choice-specific signal as defined by Frömer et al. It is also a signature that has no onset responses associated with it, given that it is thought to index motor (output) preparation. Strikingly, this motor-related signature is not considered by Frömer et al. (or noted by the reviewer), but indeed lends further evidence to the idea that our results capture the unravelling of a decision process.

In sum, we do not believe that global probe appraisal can account for the CPP, and beta dynamics observed here.

3. Decoding approach

The authors run decoding analyses using the BOLD signal, which is common practice. However, I wonder if response time is a possible confound in the analyses comparing decoding accuracies across probes and age groups (Figure 2c). For example, are decoding accuracies for one probe possibly higher because reaction times are faster than for multiple probes? Similarly, is decoding accuracy in YA higher compared to OA because their response times are faster? In these examples, longer response times might reflect greater difficulty or effort. One way to control for this possibility is to perform decoding based on beta coefficients that are extracted from a GLM that includes response times and thereby controls for difficulty or effort.

Thank you for highlighting RTs as a potential confound during the decoding analysis. As decoders were trained and tested on single time points, we argue that decoding results largely capture differences prior to probe presentation. We observe effects during the approximate period of stimulus presentation, when the probe was not yet revealed (and thus RTs may be less relevant). While we consider a full GLM single-trial decoding setup to be beyond the scope of the current manuscript (as it introduces its own modelling assumptions about how single-trial GLMs are obtained), we performed an additional control analysis based on our single-trial bootstrapped decoding estimates.

Figure R1-3. Average decoding of evidence for cued features, split between fast and slow response trials. Lines indicate periods of statistically significant differences from chance decoding accuracy (50%) as assessed by cluster-based permutation tests.

A median split (within cued attribute and uncertainty condition) of decoding accuracy between slow and fast reaction times (**Figure R1-3**) did not indicate significant differences between speed splits, while differences from chance were similarly indicated for both fast and slow response trials (see horizontal red and blue lines just beneath the 50% decoding level in Fig R1-3).

We also performed additional statistical controls: adding median reaction times of the respective conditions to the linear mixed effects models (averaging decoding accuracy across the identical time window as in the main analysis given the static nature of RT estimates) did not indicate statistically significant effects of reaction time on decoding accuracy, either across all trials ($p = 0.308$), probed trials ($p = 0.6783$), or unprobed trials ($p = 0.373$). These results jointly suggest that reaction times did not confound our decoding results.

4. Figures 4 and 5: Brainscore

It is unclear what "Brainscore" refers to in Figures 4 and 5. Brainscore seems to be the same as "linear modulation" in Figure 1, but it is unclear. As far as I can tell, the definition of brainscore is described later (Page 9) in the context of Figure 6. The figure caption suggests that theta power and pupil diameter are more strongly modulated by uncertainty in YA than OA, but it is unclear how this is related to the latent variable interpretation of brainscore, as described on page 9.

Thank you for highlighting this lack of clarity.

The reviewer correctly assumes that the display resembles the linear modulation in Figure 1, but here relates not to individual variables such as drift rates, but to a low-dimensional Brainscore. We have altered the Figure captions for Figs 4 and 5 and rewrote our Methods description (p. 20) to clarify what Brainscores are, and how they can intuitively be interpreted in the context of a task PLS.

"We obtained a summary measure of each participant- and condition-wise expression of a LV's pattern (a "Brainscore") by multiplying the vector of weights (V) by each participant's and condition's vector of input data values (P): $Brainscore = VP'$. To summarize uncertainty modulation, task PLS *Brainscores* were analyzed as described in "Linear uncertainty effect estimates"."

Brainscores project (a) multi-channel EEG loadings and (b) time-resolved pupil data into a lower-dimensional score, yielding an individual expression score for each condition and participant (see Fig. S4-1 for condition-specific Brainscores). This circumvents the need to *a priori* restrict data extraction to specific EEG channels. From these Brainscores, we quantify the linear change with increasing target load. Larger scores indeed indicate larger change in the underlying variable alongside increasing target load (theta power/pupil diameter), whereas lower values indicate dampened modulation.

Minor

5. Integration slopes

Figure 1c illustrates CPP (or integration) slopes. In the example (left plot), it was unclear to me what feature of the curves indicates the higher slopes in YA. Taking a look into the methods clarified this, but it could be highlighted in the caption by referring to the meaning of the black bar above the x-axis.

Added

6. Moreover, in the figure caption: change "slops" to "slopes" (in (c))

Corrected

7. Text S1-3 line 75: Inconsistent quotation marks ("preference")

Corrected

8. Figure 2 caption: "al" instead of "all".

Corrected

9. Decoding results (line 147-149) in the text describing significance for uncued features: Aren't color and size also partly significant, not only motion? At least, this seems to be implied by the darker lines.

This is a correct observation. The statistics indeed indicated above-chance decoding for colour and size when they were not cued. However, this pattern was temporally sparser and did not overlap with the presumed time window of

interest (stimulus presentation). The interpretation of those results is thus more challenging and may reflect false positives. We now restrict our feature-specific follow-up analysis reported in Fig. 2b to the time window of interest, defined by significant above-chance decoding accuracy of cued features (Fig 2a). This also aligns with our setup of statistical models. We have altered the caption (Fig. 2), main text (l. 160 ff.) and methods (l. 710 ff.) description accordingly.

10. Figure 4

BSR definition is explained in the caption of Figure 5 for the first time. So, seeing "BSR" in Figure 4 without definition is a little confusing.

Thank you for highlighting this discrepancy. We have added the full label to the caption.

11. Figure 4 and Discussion

The authors show pupil results, where for both YA and OA, more uncertainty is related to larger pupil size (plus the age effect). I'm not questioning this result, but I'm wondering how it relates to the literature cited in the Discussion. The authors emphasize related findings in the learning literature, such as change point detection and uncertainty-driven increases in the learning rate. These findings have in common that an agent learns more about *one* option, which is linked to larger pupil signals. For example, the agent strongly ramps up the learning rate to adjust to a change point. However, in this work, the authors see a *decrease* in the drift rate, which may be more in line with a lower learning rate and generally more cautious behavior (which is undoubtedly adaptive in the MAAT). Especially in YA, evidence accumulation about the features is distributed (i.e., concerns multiple options), akin to a lower learning rate that distributes learning across uncertain states or options. From my perspective, the authors could consider this point in the discussion of the pupil results.

Thank you for highlighting this new framing of prior work. We extended our discussion accordingly.

12. Figure S3-1 line 140: Younger "ad" change to "adults"

Corrected

13. Table S1: Some parenthesis issues, e.g., (linear mod.)

Corrected

Overall, very inspiring work!

Thank you for the encouraging comments.

Reviewer #2 (Remarks to the Author):

Kosciessa and colleagues present a rigorous examination of adult age differences in effects of “uncertainty adjustments” on perceptual attention and decision making. The study implements a recently developed multi-attribute perceptual task. The task manipulates the number of attributes that could be probed at the decision phase (i.e., “target load”; 1-4), with feature cues appearing prior to the stimulus presentation phase. Task performance was measured from the probe phase, in which one of the cued feature types is presented and participants respond based on which cued feature was represented to a greater degree. The behavioral analysis of decision process and examination of neural mechanisms is extremely rigorous, and leverages drift diffusion modeling, and analysis of EEG, fMRI and pupillometry data. Across measures, the findings converge on the conclusion that older adults show less correspondence between levels of target load and neural recruitment of resources to meet those task conditions. Novel findings in this study feature the role of the mediodorsal thalamus in supporting the assessed cognitive performance across adulthood. Despite these many strengths, the study is positioned as assessing uncertainty processing, but it’s not clear from my read that this construct was measured. The throughline of the experiment and critical findings is also hard to follow. My key concerns are outlined below, point-by-point.

We thank the reviewer for this succinct summary of our work and address individual points below.

Major comments

1. The paper needs revisions to clarify the study’s contribution to the larger literature. The authors describe the study’s key findings as highlighting “neural mechanisms whose maintenance plausibly enables flexible task set, perception, and decision computations across the adult lifespan.” On this point, I completely agree. However, the authors situate the study in the context of variations in uncertainty processing/response/management/representation/adjustment generally. In my view, the study is more accurately framed as examining how level of (or complexity in) top-down perceptual attention impacts perceptual decisions. It seems that what is referred to as “rising uncertainty” is equivalent to varying demands – specifically, attentional demands for perceptual processing. Under this interpretation, the findings that MAAT drift rates correlate with faster responses in the Incongruent Stroop condition are completely expected.

We are happy to read that the reviewer supports our reference to various psychological domains. We also note that Reviewer #3 argued for an alternative framing in the context of working memory. We are happy to clarify our chosen uncertainty framing.

We use the term “uncertainty” as a contextual challenge (Bach & Dolan, 2012), in response to which an optimal neural system needs to make multifaceted adjustments (perhaps including, but not limited to, attentional processes). This is based on the overall notion that thalamic engagement may be most sensitive to multifaceted demands (Pergola et al., 2018) under task uncertainty (Mukherjee et al., 2021). In our current research context, we consider global “attentional demands” to be insufficient to account for our findings. The reviewer (implicitly) highlights a critical difference between two different constructs: level and specificity vs. **complexity** (or **content**). Our operationalization jointly manipulates both the *size and content* of attentional sets. In our task, if it were only the *level* of attentional engagement (e.g., low-to-high engagement) that was manipulated by the set size of cues (as we read the reviewer to propose), we would also expect age-differential modulation of pre-stimulus alpha power that predicts neural and behavioral modulation during stimulus processing and beyond. This was notably not the case here (and in our previous work (Kosciessa et al., 2021)). We would also not expect differentiated decoding results that are specific to the cue *content*, i.e., features being in-/excluded in the task set.

We agree that “selective attention” is theoretically tapped in both the MAAT and the Stroop task, as irrelevant features in both tasks constitute to-be-inhibited “noise”. However, it is not clear to us why “the findings that MAAT drift rates correlate with faster responses in the Incongruent Stroop condition are completely expected”. It is only expected if behavioral metrics in both tasks sufficiently capture controlled feature *selection/suppression*. As the Stroop task never cues more than one feature, it does not allow us to investigate more varied attentional content.

Further, the general pattern of age differences for behavior is very similar to prior studies that examined age differences in top-down cognitive control tasks – as well as the general pattern of functional neuroimaging data, including the results for older adults with “young-like” functional activation responses to task conditions (e.g., Cabeza et al., 2018. Maintenance, reserve, and compensation: the cognitive neuroscience of healthy ageing. *Nature Reviews Neuroscience*, 19(11), 701-710.) The novel element in this study is the focus on perceptual attention load and decision making, which is associated with neural mechanisms not commonly observed in prior investigations of age differences attentional/EF demands (e.g., mediodorsal thalamus). This, I find very interesting and novel in the context

of cognitive aging and task demand research. In my view, this is a very strong and rigorous study that contributes novel and publishable findings, but its findings should be more directly discussed in the context of prior work on effects of cognitive task sets/demands in aging.

Thank you. We agree that our results align well with previous work on age differences in the context of multi-dimensional stimuli. We have added a dedicated discussion section on set selection costs (3.1), which we perceive to be the most relevant bridge to “top-down cognitive control” and “complexity in top-down perceptual attention” here. Notably, the relation between attention, working memory and cognitive control remains multifaceted across the lifespan (Oberauer, 2019) and beyond the scope of a concise discussion. On a side note, we argue that the multifaceted nature of computational adjustments necessitated by uncertainty is key to shed light on previously underappreciated contributions (e.g., by the mediodorsal thalamus) (Pergola et al., 2018).

The (mis)classification of cognitive function investigated has further implications. Much of the existing work on “decision making under ambiguity” involves situations where individuals do not have (and never did have) access to critical information required for an optimal decision (e.g., outcome probabilities, factual knowledge). In the current study, participants are presented with all the relevant information necessary to make a correct decision. The only uncertainty is which feature will be probed, and while participants may have certainly reflected on the level of uncertainty – the nature of the task may have made it impossible to dissociate its neural mechanisms from those of task demand or access to evidence/perceptual information during integration. If the authors disagree and believe they did isolate, and measure, the psychological construct of uncertainty, I ask that they please carefully address this in a revision.

Uncertainty can occur with regard to multiple to-be-inferred states (Bach & Dolan, 2012). The MAAT task features two external sources of uncertainty: task uncertainty and perceptual uncertainty. The latter was objectively (but not subjectively) fixed in the design and aligns with traditional operationalizations of “decision making under ambiguity” in which perceptually ambiguous input informs (concurrent) choice.

Figure S4-6. Reverse inference of uncertainty, working memory, and attention. (a-c) Consistent neuroimaging of task and state uncertainty. (a) Spatial loadings of behavioral PLS, reproduced from Fig. 6a. (b) Neuroimaging results of a state entropy manipulation with low working memory demands [reanalyzed from Fig. 1-3a in Muller, T. H., Mars, R. B., Behrens, T. E. & O'Reilly, J. X. Control of Entropy in Neural Models of Environmental State. *eLife*. 8. e39404 (2019); <https://neurovault.org/collections/4872/>]. Data range is [0 to +/- 6]. (c) Person correlation between spatial t-values and spatial loadings indicate convergent spatial activation patterns between uncertainty manipulations.

The general notion of “task demand” or “difficulty” remains computationally underspecified. We agree that our task combines multiple demands (incl. attentional selection, distractor inhibition, extra-dimensional set switching and working memory maintenance). Our neuroimaging results were consistent with a prior assessment of state uncertainty (see Fig. S4-6a-c and Text S4-6) that features low working memory load compared to our operationalization. While our work thus does not isolate “the psychological construct of uncertainty” beyond the operationalization of task uncertainty, reverse-inference analyses relate our results to prior “uncertainty” manipulations. When directly contrasting uncertainty and attention (see Fig. S4-6d), our neuroimaging results were more consistent with meta-analytic maps of uncertainty (see also our response to Reviewer 3 question 1).

Our task uncertainty manipulation coheres with similar cue conflict manipulations in the literature. In contrast to prior work in which cue conflict was manipulated via perceptually ambiguous cues (Mukherjee et al., 2021; Tsumura et al., 2021; Tsumura, Kosugi, et al., 2022; Tsumura, Shintaki, et al., 2022), our task manipulates task uncertainty via unambiguous cues that are repeated on each trial. Our design thus allows us to ask how task uncertainty impacts processing of perceptually uncertain inputs, in contrast with prior designs that ask how ambiguous task cues (operationalized via varying perceptual uncertainty) impact the processing of unambiguous inputs. However, these

cues introduce task uncertainty that is inherently *irreducible*: participants cannot use other sources of information (until the probe appears) to reduce their uncertainty about what task feature should be prioritized.

2. As I understand it, the central questions of this paper are 1) Does aging change effects feature-relevance ambiguity on goal-directed attention and decision making, and 2) If so, how? It would be most helpful if the findings were written as more direct answers to these questions. For example, it's challenging to understanding how result summaries like "Older adults express constrained uncertainty modulation of evidence integration" can be applied to the study's key questions. If I am misunderstanding the overall study objective, I hope that can be clarified.

We now use a version that is close to the reviewer's suggested questions in the framing of this work. While "feature-relevance ambiguity" is closely tied to the manipulation, we prefer the term 'task uncertainty' for simplicity, given that each of the features corresponds to a different task dimension. In prior visual discrimination studies, our manipulation of a condition in which ("subjects do not know the relevant feature to discriminate until after the offset of stimuli") has been referred to as "feature uncertainty" (Keri et al., 2004; Vogels et al., 1988). [Those studies used a block design in which feature uncertainty was either known to be present or not, thus deviating slightly from the more dynamic cue setup used here.] We consider feature uncertainty to be a narrow implementation of the more general notion of "task uncertainty" (or ambiguity) in the context of advance cues (Meiran, 2014). Task uncertainty has previously been defined as "low environmental prompts to [N.B. the input dimensions guiding] behavior" (Kray et al., 2002). Such a definition frequently features in "task switching" operationalizations that cue one of multiple sensory features as goal-relevant (e.g., the Stroop task).

3. Throughout the paper, I had some difficulty to understand exactly what was being communicated. To address this, I recommend that the language and figures should be closely reviewed and edited for clarity and simplicity.

Thank you, we have revised the text and figures with the aim to increase accessibility.

3a. Most critically, I had difficulty in understanding the central paradigm, and what exactly participants were instructed to do.

We have clarified our description of the paradigm in the methods (pg. 14 f.).

3b. I suggest that the authors make it clear exactly how uncertainty was manipulated at the beginning of the Results narrative.

Thank you for indicating this challenge. We have clarified our description of the manipulation in section 2.1:

"Stimuli were presented for three seconds, after which participants were probed as to which of the two exemplars of a single feature was most prevalent. Prior to the stimulus, participants received valid pre-stimulus cues that indicated the feature set from which a probe would be selected (with equal probability). Uncertainty was parametrically manipulated via the number of feature cues. When participants received a single cue, they could focus on a single stable feature during stimulus processing (low uncertainty); cueing of multiple features necessitated extra-dimensional attention shifts (Dias et al., 1996; Friedman & Robbins, 2022) between up to four features ("target load"; higher uncertainty) to optimally inform probe-related decisions."

3c. I had to read the text 4 times to understand the task, and I'm still not clear about how the participants' objective was described to them, other than they could earn a bonus for being correct and also quick in their decisions (online methods). Elaboration of task instructions ("Participants were told...") and edits to Figure 1a and its caption in the main manuscript would help. A revision that includes more concrete examples of different trial types would also help. Using the version of the paradigm visualization from the authors' 2021 paper would be an improvement, but naive readers may still struggle to understand how conditions map onto uncertainty. Showing at least two trial sequences with different levels of uncertainty (target load/#cues) would be beneficial.

Thank you for this suggestion. We have added depictions of example trials to the supplement (Fig. S1-o). We also clarified task instructions in the methods section ("MAAT").

3d. The paper would benefit from greater consistency in the description of the critical manipulated variables (e.g., cues, features, attributes, targets, uncertainty, probe uncertainty).

We have simplified and streamlined the used terminology:

- we consistently use *features* instead of *attributes* (with exception of the task name which we keep consistent)

- *cues* and *probes* refer to the design phases; the number of cued elements manipulates *uncertainty*
- we removed the term *probe uncertainty*
- *contextual uncertainty* was replaced by *task uncertainty*; the latter is illustrated by an example that is close to our experimental operationalization in the opening paragraph of the introduction
- we use the term *options* instead of *alternatives* to describe varying choice options for each feature.

3e. Early on, the reader needs to understand how exploration is operationalized and indexed in this study.

We removed the term “exploration” when referring to our paradigm or its results, as we do not index it in this study. We mention the term while discussing references in which exploration of alternative options or task strategies was operationalized.

3f. Usage of complex language could be paired with real-world examples. E.g., What are examples of “contexts in which feature relevance is ambiguous and require dynamic exploration”?

We now introduce a real-world example of the different cue setups in the introduction (p.3). We also simplified language in the abstract.

3g. Specialized terms or indirect language could be replaced with more specific phrases and use of lay language. E.g., for phrases like “young-like neural recruitment/signatures/modulation” and “dampened modulation of decision processes”.

We attempted to simplify the language.

Minor comments

4. The authors should provide background to justify and set expectations for pupillometry in the context of the larger study in the Intro.

We added a brief introduction to pupil dilation in the intro. We maintain a low level of detail, comparable to the other signatures. In line with Reviewer 1’s comment 11, we also expanded our discussion of pupillometry results in the context of uncertainty and aging.

5. How did the authors determine the sample sizes?

We aimed for a combined total across groups of $N = 100$ participants, with approximately age-matched sample sizes. A group size of ca. $N=50$ was based on our prior work in younger adults (Kosciessa et al., 2021, Nat. Comms.; target $N = 50$ prior to dropouts).

We have added this information to the sample description.

Reviewer #3 (Remarks to the Author):

Kosciessa and colleagues examine whether aging affects behavioral and neural signatures of uncertainty adjustment using a very well-powered cross-sectional study that includes behavior, EEG and fMRI. The paper builds on a previously published task and a number of neural correlates of the primary manipulation of that task. The idea is very interesting and the paper is a tour de force drawing on a lot of different methods using some innovative analysis techniques. The primary claim in this paper is that the aging brain is impaired in its ability to represent and deal effectively with increasing levels of uncertainty. While I find value in both the experiments performed and the claim being tested, I am not yet convinced that the data really support the claim. I have articulated my primary concerns below, and would be happy if the authors could address them through a revision.

1) Most of the claims about uncertainty here are based on sensitivity of behavior to cues in the task that indicate which (1,2,3,or 4) features could be probed at the end of the trial. This manipulation does, as authors suggest, affect uncertainty about the relevant feature dimension during information processing, but it also changes the effective working memory load of the decision – to be successful on the 4 dimension condition I would need to hold 4 feature values in working memory simultaneously and then select out the appropriate one in response to the probe. Thus, it feels like any age-differences in how behavior or EEG signals scale with uncertainty could also be interpreted in terms of differential working memory capacity, which is well established to decrease with aging.

We agree that our uncertainty operationalization requires multiple facets of working memory: while our design minimizes working memory demands for the cued task set, during increasing uncertainty requires increasing working memory load and management of working memory composition during stimulus presentation. We added a supplementary discussion regarding task uncertainty and working memory (Text S7).

However, additional analyses suggests that differential working memory capacity does not primarily account for age differences in behavior and neural engagement.

Figure R3-1 Age effects of uncertainty modulation are attenuated at high target loads.

1. Effects of constrained working memory capacity should be most pronounced at high target loads. However, age differences in neuro-behavioral modulation were most pronounced in the transition from a single-target context to a two-target context (see **Figure R3-1**), a situation in which only two solutions would have to be kept in memory. This transition is well within the capacity limits of both younger and older adults, and thus cannot be explained via differential working memory capacity. Rather, this is consistent with age differences in executive control over WM contents as observed in N-back tasks (Bopp & Verhaeghen, 2020).

Figure S4-6d. Overlap and uniqueness of uncertainty and working memory activation. Neurosynth.org activations (uniformity tests: the degree to which each voxel is consistently activated in studies that use a given term) for “working memory” and “uncertainty”, decomposed into shared and unique patterns. Both the current task uncertainty modulation and state uncertainty more closely resemble meta-analytic loadings of uncertainty than working memory or attention patterns.

2. We have performed additional reverse inference analyses to clarify the relation between our spatial fMRI pattern and prior results on uncertainty and working memory (Fig. S4-6d). When contrasting our results with prior fMRI modulation by uncertainty and working memory, our neuroimaging results were more consistent with meta-analytic maps of uncertainty (e.g., more pronounced vmPFC modulation) than working memory (e.g., more limited dlPFC modulation). This converged with prior neuroimaging results of “state uncertainty” (Muller et al., 2019), derived in a latent inference task in which working memory demands remain constant.

2) I’m not sure that the behavioral modeling really makes a tight case for uncertainty differences either (even leaving aside the point above).

Thank you for communicating this concern. We address the multiple concerns individually below.

The groups differ in their performance in the single feature condition – making it hard to say what one should expect for the higher load conditions.

We agree that behavior itself does not fully elucidate uncertainty effects. However, we performed multiple control analyses, in which we “post-hoc” subsample the behavioral data to create comparison pairs that are closely aligned in single-feature performance.

We now added an additional control analysis, in which we condition HDDM results on single-target drift rates. This analysis further indicates age-differences under uncertainty (Fig. R1-1).

We also note that age differences for single-target processing (which also link to the magnitude of uncertainty-induced modulation) argue against a working-memory account.

The model is fit flexibly to get different parameters for each condition – and does not incorporate the notion of uncertainty to make testable predictions about how performance would change across target loads.

By quantitatively comparing the fit of models that include parameter variations as a function of uncertainty (or “feature-load”) with models in which parameters are fixed across levels, we can (1) indicate the goodness of fit for models in which parameters vary alongside uncertainty. This model comparison (via DIC indices) indicates best fit for a model that includes uncertainty-dependent parameters (Fig. S1-2). This converges with the full model’s *parameter estimates*, for which uncertainty-dependent changes are consistently indicated (Fig. S1-2). We are uncertain why the reviewer assumes that this does not allow testable predictions about behavior, as the model fits are based on the prediction of RTs and accuracies (as further supported by “posterior predictions” of these behavioral metrics; Fig. S1-2b).

The interpretation of the model parameters is also muddled by the fact that participants were forced to view stimulus for 3 seconds, and thus had no need to employ a procedure for stopping evidence accumulation.

We agree that *sampling* interpretations of model parameters are challenged by the indirect link to “online” perceptual sampling. In this design, evidence accumulation does not directly index how evidence regarding multi features is sampled under uncertainty. Rather, it indirectly captures the consequences of sampling: how evidence is integrated from working memory once a probe (i.e., retro-cue) is presented. This is a fundamental constraint of the use of a retro-cue design – and a major motivation to not solely rely on behavioral or derived model data in our conclusions, but on a neuroimaging approach that can more closely provide indices of the sampling process.

Furthermore, the model fit predicts that error trials will have the same RT distribution as correct trials – which differentiates it somewhat from the memory idea posed in point 1, since failure to remember would almost certainly lead to longer RT errors. However, this aspect of the model does not seem to match the data very well... the posterior predictives in figure S1-2b suggest that participants have longer RTs than the model predicts on error trials and this is most evident for the 4 condition in older adults, precisely when you would expect memory capacity limitations to play the largest role.

We thank the reviewer for taking a close look at raw reaction time data and model-predictions. We performed additional visualizations of the RT distributions, model-predicted and empirical (Fig. R3-2). Our model indeed predicts comparable RTs for correct and incorrect responses at high target load (Fig. R3-2b), and the largest model misfit occurs for incorrect RTs when four features are cued: the model doesn't fully capture some of the slower responses. This appears to be the case for both younger and older adults, however.

Fig. R3-2. Model-predicted and observed RT distributions. (a) Predicted (lines) and empirical (shading) RT distributions, split by accuracy, target load, and age. Data are averaged across EEG and fMRI sessions. Distributions are scaled to range between 0 and 1 (see Fig. S1-2b for unscaled distributions). (b) Comparison of RT distributions to four-feature cues, split by accuracy.

The DDM is not a fully mechanistic model. As such, it does not account for individual processes such as working memory. If we understand the reviewer correctly, then similar correct and incorrect RT distributions in the four-target condition would argue against working memory failures. The implicit notion seems to be that memory failures at higher target loads are accompanied by a longer working memory search that is eventually unsuccessfully abandoned. However, if we consider gradual difficulty of memory retrieval (e.g., due to less precise feature encoding under uncertainty, or larger conflict between competing encodings), it is not clear why this would affect RT for incorrect

responses more so than accuracy more broadly. We therefore do not share the reviewer's enthusiasm that a more thorough assessment of "raw" behavioral estimates can fully disentangle these perspectives.

The imperfect model fit at higher target loads suggests that *additional* parameters (e.g., contested "collapsing boundaries") could improve empirical fits especially at high target load. However, the inter-individual relations of drift rate estimates with neuroimaging signatures indicates that our canonical parameters already capture a major factor of shared variance: EEG-based indices of evidence integration (Fig. S1-4) and fMRI modulation, pupil dilation, EEG excitability (Fig. 6).

3) The strongest behavioral evidence that the authors present toward their claim is in figure S1-3a, and while I find it much more compelling than the modeling results, I worry that there is a selection bias at play – in the face of noise in the accuracy measurements for each condition (number of targets), if we select based on one of those conditions, shouldn't we expect to see a much larger shift in the age effect for that condition? Can the authors verify that this procedure works if you use half data for selection of feature and other half for statistical testing, for example?

Figure R3-3. Cross-validated feature matching. Features are individually sorted based on single-target accuracy in the other session. n.s.: $p = 0.27$; ***: $p = 0.003$; **: $p = 0.01$; *: $p = 0.05$

We do not fully follow this rationale as accuracy should be a similarly noisy metric for any of the features as there is an equivalent number of trials for each probed feature. However, we agree that variability in feature preference could be a confound. In response to the reviewer's suggestion, we performed a *conservative* cross-validation analysis, in which we sort features based on single-target accuracy in one session (e.g., EEG) and analyze matched data in the other session (e.g., fMRI). Differences across the EEG and fMRI session environments provide a conservative test of the influence of variability in feature sorting, as the order of best-to-worst feature can systematically change across sessions. In this case, single target matching across age groups required additionally excluding the second-best feature for younger adults. Results shown in Figure R3-3 are averaged across the two possible splits. Results are consistent with those originally obtained when sorting based on data averaged across both sessions (Fig. S1-3).

4) The fMRI decoding result is interesting, but it would be nice to see whether this result actually matches the behavior. This could be done descriptively by fitting choice with a logistic regression model that contains values of relevant and irrelevant features in order to predict choice – it could also be done in the HDDM modeling framework that the authors setup by changing the bounds from reflecting (correct/incorrect) to the actual choice options and modeling the drift rate contributions of the various sources of evidence present on each trial.

We would love to replicate the fMRI decoding results with a behavioral model, but we are missing the rationale how this can be done. For every trial, the dependent variable (accuracy/RT) is only available for the single feature that was

probed. The independent predictor (evidence presented for option 1 vs. 2 for that feature) may be used to predict this for a single feature. The main constraint here is that there is no dependent behavior for features that were not probed. Regardless, we ran a logistic regression model with the available behavioral choice data:

$$\text{Choice (A vs. B)} \sim \text{evidence (A vs. B)} + \text{evidence} * \text{load} * \text{age} - \text{load} - \text{age} - \text{age} * \text{load} + (1|id) + (\text{load}|id)$$

We ran independent models, in which evidence either referred to the probed feature, or to evidence related to either of the four individual features, regardless of whether they ended up being the probe (i.e., *probe-agnostic*). Note that these predictors are not orthogonal given that a feature will be relevant 1/4th of the time. In addition, there will be an option overlap some of the time given that there are only two possible response mappings (see Kosciessa et al., 2021 for a related control analysis).

Fig. R3-4a. Logistic regression results. Coefficients and p-values from logistic regression models in which choice (choice A vs. B) is predicted from presented evidence (favouring option A vs. B).

For the model with probed feature evidence as the predictor (= fMRI decoding model), we replicate significant main effects of evidence ($p \sim 0$), an evidence x age interaction ($p \sim 0$), and an evidence x load interaction ($p \sim 0$), but no three-way interaction ($p = 0.21$). Effects were attenuated and less robust for probe-agnostic models (Fig. R3-4a). In short, we can behaviorally replicate a decreasing predictability of probed evidence (*which converges with how choice accuracy is defined**), but behavioral estimates do not allow us to test increased predictability of ultimately unprobed features.

**For the probed feature, accuracy is defined as the proportion of choices that aligned with the shown evidence. Logistic regression models add to this picture by specifying whether there was potential bias towards either alternative.*

As a minor point it would be useful if fMRI decoding plots were on same axis for older/younger adults.

Fig. R3-4b. Decoding by age and target load. Means \pm within-subject SEM. Blue = young adults. Red = older adults.

We have aligned axes in Fig. 2c and added a distribution plot for the age effect. If the reviewer refers to age-split data being presented as a function of target load (despite the lack of indicated interaction effect), we show these data in Fig. R3-4b.

5) Pupil data shows a very strong trend consistent with stronger conditional modulation in young adults. However, if I understand the task correctly, there is a luminance difference in the conditions that occurs 2 seconds before stimulus onset – is it possible that some of the observed changes are carryover effects from this luminance difference?

Figure S4-2. Pupil size modulation in response to cue onset. (a) Temporal derivatives of pupil diameter linearly decreased with increasing number of feature cues. Data are means \pm within-subject S.E.M. across age groups. The black line indicates significant loadings of a Task PLS LV1 (*permuted* $p \sim 0$). The inset shows grand average pupil dynamics with shaded cue and stimulus phases. (b) Relative pupil constriction upon presentation of more cues (= increasing luminance) did not differ between age groups, whereas younger adults showed larger relative pupil size for (luminance-matched) stimuli in the face of more uncertain feature relevance than older adults.

The reviewer correctly notes that the cue presented *three* seconds prior to stimulus onset varies in luminance given that one to four bright cues are presented. We have therefore performed additional analyses, reported in Text S4-2 and Figures S4-2. These analyses jointly indicate that stimulus-related age differences in uncertainty-related pupil size modulation (shown in Fig. 4b) are not merely a spill-over of differential luminance sensitivity.

6) Figure 1C: I found it hard to understand the link between the averages on the left and the per-subject quantifications on the right, since the difference between 1 and 4 in the older adult averaged data looks just as extreme as the one for younger adults – but the quantification suggests a massive difference.

The individual estimates indeed indicate substantial reductions in drift rate as target load increases. Individual data by target load are shown separately for each age group in the insets in Fig. S1-4. Individual data from the insets also highlights more clearly the attenuated profile of slope changes under target load in older adults. On a minor, but related note: the averages on the left are scaled to identical amplitudes at the time of choice. Unscaled data are shown in Fig. S1-4.

7) The whole brain analysis linking behavioral signatures of uncertainty reduction to specific patterns of uncertainty-related brain activation seems like one of the most interesting parts of this paper, yet I found the actual procedure a bit difficult to follow given that the different parts of the analysis were described in different sections of the methods and very little intuition was provided for the motivations of the exact procedures in the results. Beyond this basic lack of clarity, I was a bit concerned that the brain activations pulled out by this complex analysis might simply just be the pattern of brain activations to the manipulation of interest – rather than saying something more specific.

We have edited the behavioral PLS description (p. 20).

The reviewer correctly assumes that this *inter-individual difference* analysis (Fig. 6) captures a similar brain activation pattern as observed for the uncertainty manipulation itself (cf. LV1 in Fig. S6-1a). We now note this on pg. 9. This supports the notion that the multi-modal model captures inter-individual differences in adjustment to uncertainty (cf. task PLS) across signatures ranging from behavior over EEG and pupil to fMRI BOLD. This is not trivial: the behavioral PLS cannot recapture the loadings of the task PLS *unless* the “behavioral” variables relate to the same condition modulation. Task PLS exclusively captures condition separation at the group level, whereas behavioral PLS is sensitive to between-personal correlations across variables. As such, the appearance of similarity after the fact is not trivial.

8) In the schematic/summary figure, the link to the attractor network changes described feel a bit tenuous to me. The attractors depicted are point attractors, whereas the behavioral data are all interpreted through DDM models, which have internal dynamics consistent with a line attractor. The point attractor idea also seems very similar to proposals about prefrontal working memory representations, which again makes me wonder how different the effects reported here are from things that have been reported with respect to age-related changes in working memory capacity and its neural correlates.

Thank you for the detailed comment. The point attractors in the schematic cartoon merely aim to indicate the fidelity of separating different feature representations. We do not propose an explicit model of corresponding (potentially attractor-based) dynamics. We have therefore removed the term “attractor” from the figure. In the pictorial depiction, the troughs in the dynamic landscape are intended to illustrate different feature representations during stimulus presentation, as opposed to the subsequent probe-related sampling (which is captured by behavior/DDM). We added this information to the legend of Fig. 7.

References

- Bach, D. R., & Dolan, R. J. (2012). Knowing how much you don't know: a neural organization of uncertainty estimates. *Nat Rev Neurosci*, 13(8), 572-586. <https://doi.org/10.1038/nrn3289>
- Bopp, K. L., & Verhaeghen, P. (2020). Aging and n-Back Performance: A Meta-Analysis. *J Gerontol B Psychol Sci Soc Sci*, 75(2), 229-240. <https://doi.org/10.1093/geronb/gby024>
- Dias, R., Robbins, T. W., & Roberts, A. C. (1996). Dissociation in prefrontal cortex of affective and attentional shifts. *Nature*, 380(6569), 69-72. <https://doi.org/10.1038/380069a0>
- Friedman, N. P., & Robbins, T. W. (2022). The role of prefrontal cortex in cognitive control and executive function. *Neuropsychopharmacology*, 47(1), 72-89. <https://doi.org/10.1038/s41386-021-01132-0>
- Keri, S., Decety, J., Roland, P. E., & Gulyas, B. (2004). Feature uncertainty activates anterior cingulate cortex. *Hum Brain Mapp*, 21(1), 26-33. <https://doi.org/10.1002/hbm.10150>
- Kosciessa, J. Q., Lindenberger, U., & Garrett, D. D. (2021). Thalamocortical excitability modulation guides human perception under uncertainty. *Nature Communications*, 12(1), 2430. <https://doi.org/10.1038/s41467-021-22511-7>
- Kray, J., Li, K. Z., & Lindenberger, U. (2002). Age-related changes in task-switching components: the role of task uncertainty. *Brain Cogn*, 49(3), 363-381. <http://www.ncbi.nlm.nih.gov/pubmed/12139959>
- Meiran, N. (2014). The task-cuing paradigm: A user's guide. In *Task switching and cognitive control*. (pp. 45-73). Oxford University Press. <https://doi.org/10.1093/acprof:osobl/9780199921959.003.0003>
- Mukherjee, A., Lam, N. H., Wimmer, R. D., & Halassa, M. M. (2021). Thalamic circuits for independent control of prefrontal signal and noise. *Nature*. <https://doi.org/10.1038/s41586-021-04056-3>
- Muller, T. H., Mars, R. B., Behrens, T. E., & O'Reilly, J. X. (2019). Control of entropy in neural models of environmental state. *Elife*, 8. <https://doi.org/10.7554/eLife.39404>
- Oberauer, K. (2019). Working Memory and Attention - A Conceptual Analysis and Review. *J Cogn*, 2(1), 36. <https://doi.org/10.5334/joc.58>
- Pergola, G., Danet, L., Pitel, A. L., Carlesimo, G. A., Segobin, S., Pariente, J., Suchan, B., Mitchell, A. S., & Barbeau, E. J. (2018). The Regulatory Role of the Human Mediodorsal Thalamus. *Trends Cogn Sci*, 22(11), 1011-1025. <https://doi.org/10.1016/j.tics.2018.08.006>
- Tsumura, K., Aoki, R., Takeda, M., Nakahara, K., & Jimura, K. (2021). Cross-Hemispheric Complementary Prefrontal Mechanisms during Task Switching under Perceptual Uncertainty. *J Neurosci*, 41(10), 2197-2213. <https://doi.org/10.1523/JNEUROSCI.2096-20.2021>
- Tsumura, K., Kosugi, K., Hattori, Y., Aoki, R., Takeda, M., Chikazoe, J., Nakahara, K., & Jimura, K. (2022). Reversible Fronto-occipitotemporal Signaling Complements Task Encoding and Switching under Ambiguous Cues. *Cereb Cortex*, 32(9), 1911-1931. <https://doi.org/10.1093/cercor/bhab324>
- Tsumura, K., Shintaki, R., Takeda, M., Chikazoe, J., Nakahara, K., & Jimura, K. (2022). Perceptual Uncertainty Alternates Top-down and Bottom-up Fronto-Temporal Network Signaling during Response Inhibition. *J Neurosci*, 42(22), 4567-4579. <https://doi.org/10.1523/JNEUROSCI.2537-21.2022>
- Vogels, R., Eeckhout, H., & Orban, G. A. (1988). The effect of feature uncertainty on spatial discriminations. *Perception*, 17(5), 565-577. <https://doi.org/10.1068/p170565>

REVIEWER COMMENTS

Reviewer #1 (Remarks to the Author):

Review Summary

Thank you for the revision of the manuscript. My comments about age-matched target accuracies (#1), the potential RT confound in the decoding analyses (#3), and the “brain score” explanation (#4) have been adequately addressed. However, my concern about the CPP interpretation (#2) remains.

The authors highlight that appraisal of the probe set is unlikely to explain the CPP effect, but from my perspective, they don't convincingly rule out the potential confound due to stimulus-locked activity. The authors mention Froemer et al.'s point concerning the domain of perceptual decision-making, and argue against it. Their explanation for why an evidence accumulation perspective can remain valid even if response-locked ramping can be explained by stimulus-locked potentials has some plausibility.

However, on the other hand, and as the authors mention as well, the evidence accumulation signal should be closely related to RT, and the peak of the signal would be expected to occur earlier for faster compared to slower responses. Therefore, removing stimulus-related activity from the response-related evidence accumulation signal should not necessarily eliminate pre-response ramping, even though evidence accumulation takes place quickly after stimulus onset.

Therefore, I still argue that the authors should test directly whether deconvolving stimulus-related activity changes their CPP results.

To be clear, even if the CPP results of the present paper change after deconvolution, it would absolutely not be a reason for me to suggest a rejection of the paper. However, I think it is important to take this emerging perspective on the CPP and evidence accumulation seriously. If deconvolution of stimulus-related activity eliminates the CPP, the authors could discuss why they would nevertheless assume the presence of a neural evidence accumulation process.

Reviewer #2 (Remarks to the Author):

I thank the authors for their thorough responses. Added clarifications, specifications, linkages to current literature on cognition and cognitive aging, and real-world examples have enhanced the paper and I find it's ready for publication. Congrats to the authors on this work. I look forward to following their future extensions of this research.

Reviewer #3 (Remarks to the Author):

The authors have fully addressed my concerns.

Response to reviewers

We thank the reviewers for their evaluation of our first revision.

Reviewer #1 (Remarks to the Author):

Review Summary

Thank you for the revision of the manuscript. My comments about age-matched target accuracies (#1), the potential RT confound in the decoding analyses (#3), and the “brain score” explanation (#4) have been adequately addressed. However, my concern about the CPP interpretation (#2) remains.

The authors highlight that appraisal of the probe set is unlikely to explain the CPP effect, but from my perspective, they don't convincingly rule out the potential confound due to stimulus-locked activity. The authors mention Froemer et al.'s point concerning the domain of perceptual decision-making, and argue against it. Their explanation for why an evidence accumulation perspective can remain valid even if response-locked ramping can be explained by stimulus-locked potentials has some plausibility.

However, on the other hand, and as the authors mention as well, the evidence accumulation signal should be closely related to RT, and the peak of the signal would be expected to occur earlier for faster compared to slower responses. Therefore, removing stimulus-related activity from the response-related evidence accumulation signal should not necessarily eliminate pre-response ramping, even though evidence accumulation takes place quickly after stimulus onset.

Therefore, I still argue that the authors should test directly whether deconvolving stimulus-related activity changes their CPP results.

To be clear, even if the CPP results of the present paper change after deconvolution, it would absolutely not be a reason for me to suggest a rejection of the paper. However, I think it is important to take this emerging perspective on the CPP and evidence accumulation seriously. If deconvolution of stimulus-related activity eliminates the CPP, the authors could discuss why they would nevertheless assume the presence of a neural evidence accumulation process.

We agree that – while not critical to the main arguments of our manuscript – a deconvolution analysis can inform the emerging discussion on putative EEG correlates of evidence accumulation. Based on the raised concern about the impact of probe presentation on response-locked CPPs, our second revision adds a supplementary analysis of deconvolved response-locked ERPs. This analysis closely follows the procedures reported in the preprint by Frömer et al. to allow direct comparison. Results are briefly referenced in the main text in lines 139 ff.: “Reduced modulation of pre-response slopes in older adults was observed (at both central and parietal sites) also after controlling for overlapping potentials locked to probe onset (Text S1-7).” We replicate the report of this analysis in blue below.

The deconvolved results are consistent with our main result of attenuated uncertainty modulation of pre-response EEG potentials in older adults, despite notable changes in CPP topography and morphology of the potential closer in time to probe presentation. This is expected following the statistical removal of probe-locked activation and indicates a partial dissociation between stimulus- and response-related potentials. We discuss the methodological difficulty of disambiguating two possible accounts: one in which a single latent choice process covaries across the two modelled time periods, and the alternative of (partially) separable appraisal and choice processes. We further argue, also based on preserved ramping post-deconvolution at central sites, that such concerns about probe-related potentials are less convincing for effector-specific (here: motor) signatures. We would also like to highlight that our central argument regarding age differences in the modulation of evidence accumulation is not primarily dependent on EEG signatures but is critically evidenced by HDDM analyses that directly model RT and accuracy without reference to temporally overlapping EEG potentials. We hope that these datapoints and arguments can broaden the discussion on the interpretation of the CPP in the context of perceptual decision-making.

Supplementary Fig. S1-7. Deconvolution analyses for response-locked potentials. (a) Average peri-response topographies from -250 ms to 100 ms before (left, cf. Fig. 1c) and after (right) probe onset deconvolution. Black stars indicate sensors selected for parietal/CPP potentials, whereas white stars indicate central channels selected for motor-related potentials. (b) Response-related potentials before (left) and after (right) probe onset deconvolution. Means \pm within-subject SEM. Parietal (top) and central (bottom) potentials correspond to the channels indicated in a. (c) Uncertainty modulation (linear) of pre-response slopes (deconvolved data; cf. Fig. 1c) by age group. Pre-response slopes were estimated during the time windows indicated by the shaded areas in b.

Supplementary Text S1-7. Deconvolution of probe-related from response-locked ERP potentials.

Recent work proposes that pre-response CPP ramping could be sufficiently explained by a probe-related appraisal process – rather than choice – that is jittered in time due to response speed variations (Frömer et al., 2024). To explore this here, we conducted deconvolution analyses using the unfold toolbox (Ehinger & Dimigen, 2019) implemented in MATLAB. By providing event-specific regression formulas, this approach can disentangle temporally overlapping ERP responses. We jointly modelled probe onset and response regressors using FIR basis functions, time expanded ± 1 seconds around the respective events (Frömer et al., 2024). Data was 8 Hz lowpass-filtered, no baseline correction was applied, amplitudes $> 250 \mu V$ were removed. Consistent with the CPP reflecting a stimulus-related P300 potential (O'Connell et al., 2012; Twomey et al., 2015), posterior topographies changed following removal of onset-related activity (Fig. S1-7). However, parieto-occipital potentials more exhibited residual ramping that peaked slightly prior to response onset, in line with a choice-related process. Age differences in the uncertainty modulation of residual slopes (linear fits: -500 to -50 ms) resembled those in the main analysis (cf. Fig. 1c), albeit with no uncertainty-related slope adjustment observed in older adults. Analogous to beta power dynamics (Fig. S1-6), a central motor-related potential indicated that uncertainty shallowed slopes of the deconvolved motor negativity (linear fits: -500 to -0 ms).

While onset deconvolution captures the impact of probe processing on response-locked potentials, it provides inconclusive information regarding an accumulation-to-bound decision process. Differences in evidence integration rate can emerge rapidly after probe onset, persist until response (see e.g., Churchland et al., 2008; Twomey et al., 2015), and produce differential RT distributions (Twomey et al., 2015). In the face of temporal covariance, considering unique response-locked variance would remove substantial “choice” signal of interest. In the face of this ambiguity regarding the latent processes contributing to observed potentials, multiple observations support a choice-related account here: First, CPP slopes inter-individually mirror model-based

(HDDM) estimates of evidence integration (Figure S1-5c). Second, similar ramping dynamics are observed in the motor system (Text S1-6, central ERP above), where probe impact is more limited. Notably, contralateral beta power reflects relative biases towards one hemispheric response over the other, thereby fulfilling strict definitions of a choice-specific signature (Frömer et al., 2024). Effector-specific signatures (Donner et al., 2009) therefore lend further credence to the notion that our neural signatures capture evolving decision processes.

Reviewer #2 (Remarks to the Author):

I thank the authors for their thorough responses. Added clarifications, specifications, linkages to current literature on cognition and cognitive aging, and real-world examples have enhanced the paper and I find it's ready for publication. Congrats to the authors on this work. I look forward to following their future extensions of this research.

We again thank the reviewer for the constructive feedback.

Reviewer #3 (Remarks to the Author):

The authors have fully addressed my concerns.

We again thank the reviewer for the constructive feedback.

References

- Churchland, A. K., Kiani, R., & Shadlen, M. N. (2008). Decision-making with multiple alternatives. *Nat Neurosci*, 11(6), 693-702. <https://doi.org/10.1038/nn.2123>
- Donner, T. H., Siegel, M., Fries, P., & Engel, A. K. (2009). Buildup of choice-predictive activity in human motor cortex during perceptual decision making. *Curr Biol*, 19(18), 1581-1585. <https://doi.org/10.1016/j.cub.2009.07.066>
- Ehinger, B. V., & Dimigen, O. (2019). Unfold: an integrated toolbox for overlap correction, non-linear modeling, and regression-based EEG analysis. *PeerJ*, 7, e7838. <https://doi.org/10.7717/peerj.7838>
- Frömer, R., Nassar, M. R., Ehinger, B. V., & Shenhav, A. (2024). Common neural choice signals can emerge artifactually amidst multiple distinct value signals. *bioRxiv*, 2022.2008.2002.502393. <https://doi.org/10.1101/2022.08.02.502393>
- O'Connell, R. G., Dockree, P. M., & Kelly, S. P. (2012). A supramodal accumulation-to-bound signal that determines perceptual decisions in humans. *Nat Neurosci*, 15(12), 1729-1735. <https://doi.org/10.1038/nn.3248>
- Twomey, D. M., Murphy, P. R., Kelly, S. P., & O'Connell, R. G. (2015). The classic P300 encodes a build-to-threshold decision variable. *Eur J Neurosci*, 42(1), 1636-1643. <https://doi.org/10.1111/ejn.12936>

REVIEWERS' COMMENTS

Reviewer #1 (Remarks to the Author):

I thank the authors for additionally performing the deconvolution analysis. My concern about the CPP has been fully addressed. I think the control analysis makes this point of the paper more convincing and, more generally, contributes to the ongoing debate about the CPP as an evidence accumulation signal. Congrats to the authors on this great paper!